# A porin-like protein used by bacterial predators defines a wider lipid-trapping superfamily

Rebecca J. Parr[1,8], Yoann G. Santin[2,10], Giedrė Ratkevičiūtė[1,9,10], Simon G. Caulton [1], Paul Radford [3], Dominik Gurvič[4], Matthew Jenkins [1], Matthew T. Doyle [5], Liam Mead[1], Augustinas Silale [6], Bert van den Berg [6], Timothy J. Knowles [1], R. Elizabeth Sockett [3], Phillip J. Stansfeld [4], Géraldine Laloux [2,7] ✉ & Andrew L. Lovering [1] ✉

Outer membrane proteins (OMPs) define the surface biology of Gram-negative bacteria, with roles in adhesion, transport, catalysis and signalling. Specifically, porin beta-barrels are common diffusion channels, predominantly monomeric/trimeric in nature. Here we show that the major OMP of the bacterial predator *Bdellovibrio bacteriovorus*, PopA, differs from this architecture, forming a pentameric porin-like superstructure. Our X-ray and cryo-EM structures reveal a bowl-shape composite outer β-wall, which houses a central chamber that encloses a section of the lipid bilayer. We demonstrate that PopA, reported to insert into prey inner membrane, causes defects when directed into *Escherichia coli* membranes. We discover widespread PopA homologues, including likely tetramers and hexamers, that retain the lipid chamber; a similar chamber is formed by an unrelated smaller closed-barrel family, implicating this as a general feature. Our work thus defines oligomeric OMP superfamilies, whose deviation from prior structures requires us to revisit existing membrane-interaction motifs and folding models.

Outer membrane proteins (OMPs) have major roles in bacterial and organelle biology, with classical beta-barrel structures following accepted common rules and folding pathways[1]. Proteins in the outer membrane (OM) are dominantly composed of β-strands, and those in the inner membrane (IM) α-helical; one anomaly to this preference comes from rare inter-membrane transfer events, such as the translocation of the *Neisseria* OMP/porin PorB into mitochondria during pathogenesis[2] or the transfer of an OMP from the bacterial predator *Bdellovibrio bacteriovorus* into the IM of the prey cell[3,4]. PorB has been

characterized, adopting a consensus porin fold[2]; porins are chiefly monomeric or trimeric barrels of low substrate specificity[5]. *B. bacteriovorus* is a model predatory bacterium, whose ability to enter and kill a wide range of other Gram-negative bacteria is of interest to the study of staged lifecycles and bacterial cell physiology, but also of potential use in several applied contexts[6]. Prey membrane function and manipulation is key to *Bdellovibrio* predation, factoring during cell entry and exit, and potentially nutrient transfer[7]. *Bdellovibrio* synthesizes an unusual neutral lipid A[8], but retains identifiable OM Bam, Lpt and Mla transport

[1]School of Biosciences, University of Birmingham, Birmingham, UK. [2]de Duve Institute, UCLouvain, 75 Avenue Hippocrate 75, bte B1.75.08, Brussels, Belgium. [3]School of Life Sciences, Nottingham University, Medical School, Queen's Medical Centre, Nottingham, UK. [4]School of Life Sciences & Department of Chemistry, Gibbet Hill Campus, The University of Warwick, Coventry, UK. [5]School of Medical Sciences, Faculty of Medicine and Health, The University of Sydney, Darlington, NSW, Australia. [6]Biosciences Institute, Faculty of Medical Sciences, Newcastle University, Framlington Place, Newcastle upon Tyne, UK. [7]WEL Research Institute, 6 Avenue Pasteur, Wavre, Leuven, Belgium. [8]Present address: MPL, Diamond Synchrotron, Didcot, UK. [9]Present address: Department of Biochemistry, University of Oxford, Dorothy Crowfoot Hodgkin Building, South Parks Road, Oxford, UK. [10]These authors contributed equally: Yoann G. Santin, Giedrė Ratkevičiūtė. ✉e-mail: geraldine.laloux@uclouvain.be; a.lovering@bham.ac.uk

machineries[9]. Two independent studies reported the transfer of a *Bdellovibrio* OMP into the inner membrane of prey[3,4] – later determined to be encoded by the *bd0427* gene[10]. Ficoll-separated OM fractions revealed that Bd0427 is the major OMP of HD100, and its homologues are likewise the major component in other predatory strains[11].

Here we discover that Bd0427 defines a distinct class of oligomeric porin-like proteins, differing from the hundreds of characterised Gram-negative trimeric barrels by adopting a pentameric state. Our x-ray and cryo-EM structures reveal a plugged lumen and a central chamber that forms from a "folding-down" of the barrel wall and is thus able to trap a lipid monolayer. We use molecular dynamics simulations to visualize interactions with the membrane and demonstrate that recombinant expression in *E. coli* prey leads to IM defects. Structure-based searches identify homologues in both predatory and non-predatory species, and establish Bd0427 (which we rename PopA) as the exemplar of a distinct class of membrane proteins.

## Results

### PopA adopts a pentameric assembly distinct from other OMPs

We were able to produce and refold Bd0427 (aa 21-353) using dropwise dilution of 8 M urea-solubilized inclusion material into a buffer containing either β-octylglucoside (βOG) or dodecylmaltoside (DDM) detergents. Refolding, as judged by heat-modifiability (Supplementary Fig. 1), allowed purification and generation of several crystal forms; herein we focus on the best-diffracting C222₁ form (2.8 Å resolution, statistics provided in Supplementary Table 1). Representative electron density from several key regions of the fold is provided in Supplementary Fig. 2. There are five Bd0427 copies in the asymmetric unit, forming an unprecedented pentameric OMP; from hereon in, we rename Bd0427 as PopA (<u>p</u>entameric <u>o</u>uter membrane <u>p</u>rotein <u>A</u>).

PopA forms a 16-stranded antiparallel β-barrel, with the lumen plugged by an alpha-helical domain, as shown in Fig. 1a,b,c. Distinct from a regular cylinder of uniform height, strands 1–4 and 14–16 are shorter, creating a "flattened edge" 20 Å high on the inward side, considerably shorter than the 55 Å outer rim. PopA retains some features present in "textbook" OMPs, with a hydrophobic band of residues ~25 Å wide for interaction with the bilayer, and identifiable aromatic girdles (Fig. 1d) known to delineate the membrane interface[5]. PopA has weak structural similarity to several experimentally-determined proteins, including FadL[12] (3.3 Å rmsd, Cα) and OmpT[13] (3.6 Å rmsd). Like FadL, PopA has a plug and disrupted hydrogen-bonding between barrel strands (but at a different relative location, Supplementary Fig. 3) and has two more β-strands in its fold. PopA also differs from trimeric porins by using seams at its edges to form a continuous outer rim (Fig. 1g). Hence, the PopA fold is divergent from previously characterised OMPs.

The pentameric assembly is radially organized around loop 8, a folded-over edge of the barrel, making a chamber at the centre of the oligomer (Fig. 2c–e). The net effect of oligomerisation creates a ~100 Å wide bowl-like architecture (rather than five grouped cylinders), whose outer rim is formed from a continuous "wall" of strands 5–14 (Fig. 1c); this is also distinct from classical porins[5]. The extracellular face of the pentamer is strongly negatively charged (Fig. 1c) with a notable ring of lysines beneath (Fig. 1e). In comparison to classical 16-stranded trimeric porins, the closest relative to PopA is an anion transporter from *Comamonas acidovorans* (PDB 1E54[14], Foldseek[15] score 4.8 e-5, sequence identity 13%); superposition reveals that the PopA fold has an altered register such that β-strand order differs by 2 *e.g.*, PopA strand 12 sits over trimeric porin strand 14. This has the outcome of roughly aligning two chains of the PopA pentamer with classical porins, with the central chamber replacing the third.

The FadL-like plug (Fig. 2a) is held in place by polar and nonpolar interactions (Consurf[16] analysis indicates that the plug is highly-conserved, Supplementary Fig. 4 and 5). The three small α-helices of the plug, along with R32 and R43, prevent any potential channel through the barrel of PopA. This closure can be quantified by analysis with MOLE[17], which confirms a constriction point of 0 Å (Fig. 2b).

### PopA assembles around a central, trapped lipid monolayer

The pentamer has a central chamber formed by the association of five loop 8 peptides that is open to the periplasmic side of the bilayer (Fig. 2c); the chamber is effectively shaped to the dimensions of a lipid monolayer. Consistent with this, the inside of the chamber is lined with hydrophobic residues contributed by the barrel surface and lower face of loop 8 (Fig. 2d, e). During model building, it became apparent that this feature exhibited strong electron density, into which we could model ten octane molecules (Fig. 2f, g), representative of the tail of βOG detergent. There is no apparent channel for lipids to move laterally into or out of this central chamber. A second region of difference density was located behind the α-helix of loop 2 (Supplementary Fig. 6). While using the Alphafold3 server[18] to replicate fatty acid binding at the central chamber, we noticed it would additionally place palmitate within this density. Refinement verified that palmitate fits the density well (Fig. 2h), with the carboxylate contacting the side chain of the conserved R88 and the acyl chain situated opposite the bulge created by the disrupted β-strands around S189 that opens to the bilayer. Fatty acid binding adds to the features (general fold, β-bulge, plug domain) shared between PopA and FadL.

### Cryo-EM and molecular dynamics simulations confirm PopA monolayer-trapping features

Having observed our original structural information from refolded PopA, we set out to interrogate the native folded state, using our existing structure to inform on introducing an internal His-tag in loop 2, referred to hereafter as PopA_his. The protein was successfully solubilized in 1% (w/v) DDM, and was amenable to cryo-EM analysis. A final particle set of 365,835 was subjected to ab initio reconstitution followed by non-uniform refinement in C1 symmetry. The resultant map at 2.85 Å resolution (according to the gold-standard Fourier shell correlation curve with a threshold of 0.143, Supplementary Figs. 7, 8) allowed building of a model with correlation coefficient (CC mask) of 0.87 (Supplementary Table 2).

The cryo-EM structure of PopA_his retains the five-fold arrangement of the X-ray models, demonstrating that differences in native:refolding methodology and DDM:βOG usage have no significant outcome on the PopA structure (Fig. 3a, chain rmsd to X-ray ranging from 0.7–1.1 Å). The central chamber in PopA_his also has density for alkyl chains, equivalent to 11 DDM molecules, two per monomer with an additional one at the centre (Fig. 3a).

We subsequently performed multi-scale molecular dynamics simulations of PopA in both asymmetrical OM and symmetrical IM models, as both a monomer and pentamer (Fig. 3b, c). Based on the experimental density, we inserted five phosphatidylethanolamine lipids into the central chamber and calculated the simulated densities over three repeats of 500 ns MD simulations. The simulated density agrees favourably with the densities observed from the structural data. As a monomer, PopA shows further fluctuations in the extracellular loops, illustrating that the individual monomers are stabilised in the pentamer complex (Supplementary Fig. 9–11). The cryo-EM structure also retains density behind the helix of loop 2 (Supplementary Fig. 6b), and we were able to validate the x-ray and cryo-EM attribution of this to palmitate (and thus potential FadL-like role) by demonstrating high contact residency time for this ligand during simulations (Fig. 3d).

### PopA is not a porin but can induce defects in *E. coli* prey membranes

We undertook transport assays to monitor if PopA demonstrates any specificity for size or class of substrate. We utilised liposome swelling assays, comparing swelling rates to those of proteoliposomes containing the well-characterized *E. coli* OmpF porin[19]. Relative to

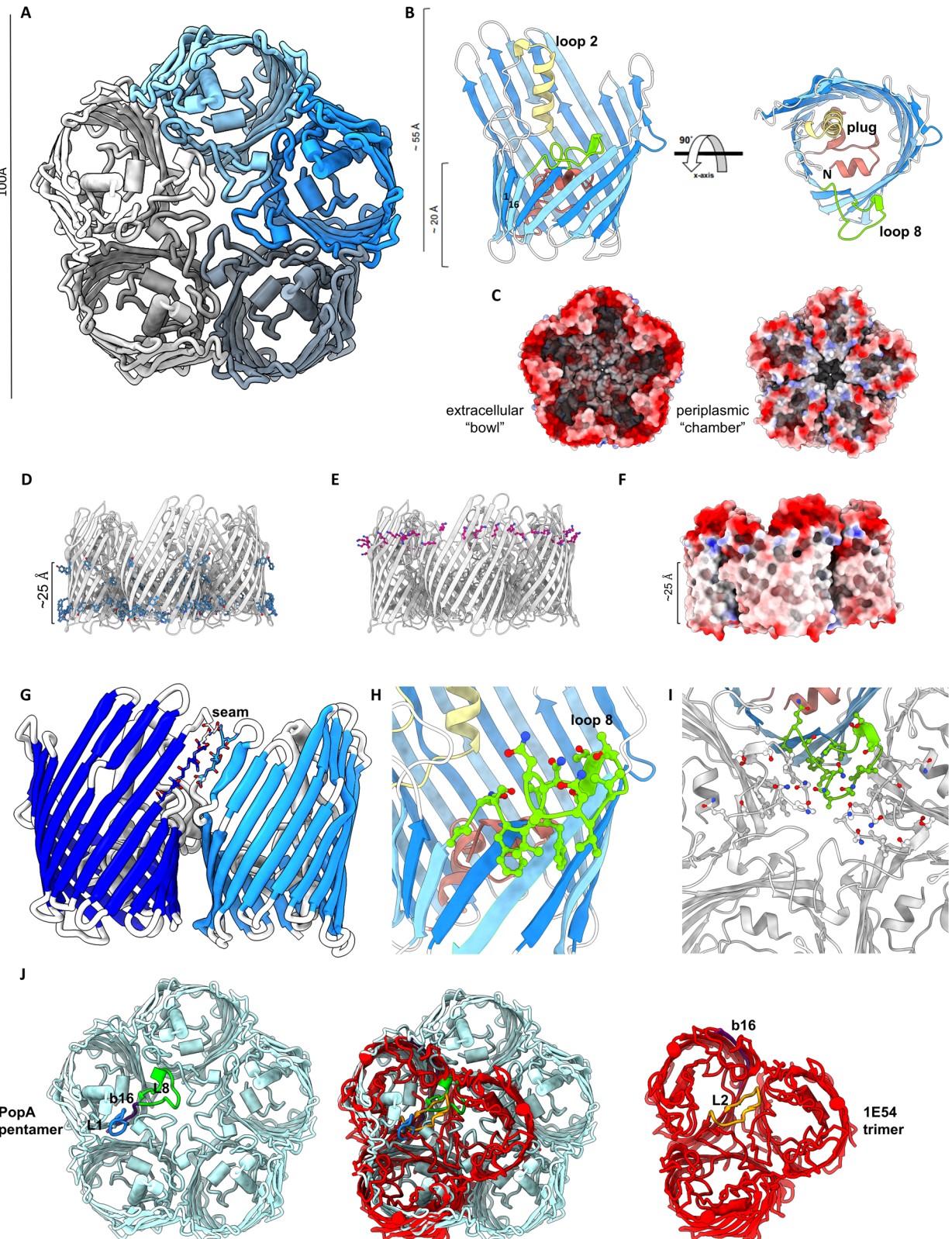

**Fig. 1 | PopA is a divergent OMP Pentamer. A** Ribbon diagram of the PopA pentamer. **B** Two views highlighting the wall-reinforcing alpha helix of loop 2 (pale yellow), plug (pink) and oligomerisation loop 8 (green, forms centre of structure shown in panel A). **C** Electrostatic representation of PopA pentamer faces (extracellular left, periplasmic right). **D** Aromatic girdle residues (blue sticks), and inferred membrane spacing. **E** Lysine ring (purple sticks) at extracellular face. **F** Hydrophobic span (white) of PopA transmembrane β-strands. **G** Monomers make a continuous outer sheet wall via seams (backbone shown in stickform) **H, I** Residue sidechain detail for loop 8 and central chamber. **J** Oligomer comparison (overlay, centre) of PopA (blue, left) to trimeric anion transporter (1E54[14], red, right), demonstrating arrangement around different loops (coloured individually, strand β16 of both is purple to provide a barrel orientation guide).

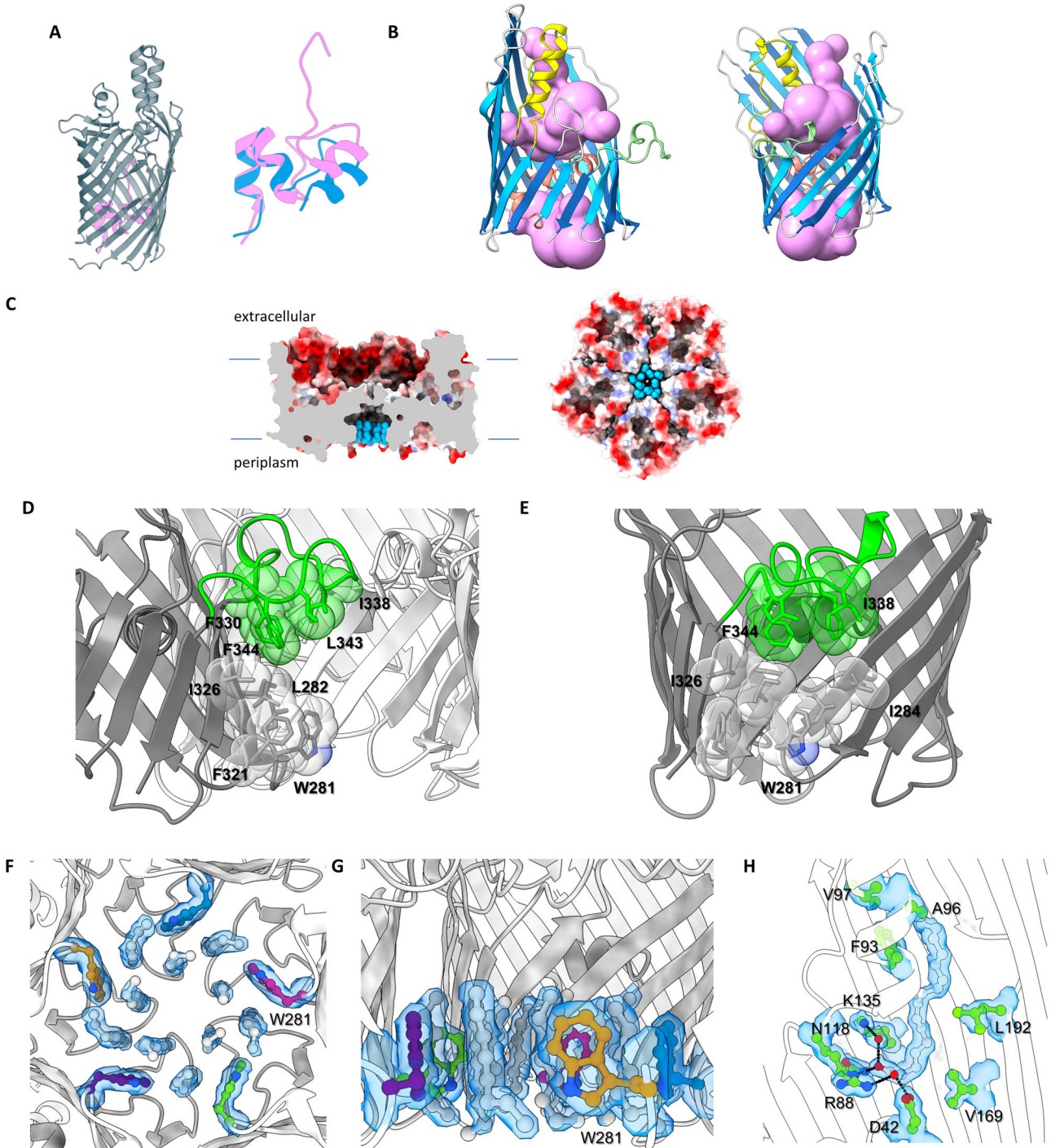

**Fig. 2 | Pore and membrane features of PopA. A** Lefthand, ribbon diagram of FadL with plug (pink). Righthand, overlay of PopA plug (blue). **B** Lack of pore lumen in PopA, showing constriction of solvent-accessible volume (pink) as calculated using MOLE[17]. **C** Side-on and bottom views of PopA pentamer, demonstrative that the central chamber is located at the phospholipid/periplasmic side of the asymmetric outer membrane bilayer. **D** (side view), **E** (end-on) Spacefill of hydrophobic residues lining central chamber (from one monomer), using cutaway-view omitting one chain. **F** (from periplasm), **G** (in plane of membrane) Electron density and fit of alkyl chains in central chamber, with surrounding aromatic residues, 2fo-fc map at 1σ. **H** Fit of bound palmitate (white) under sidewall helix, 2fo-fc map at 1σ. The carboxylate of palmitate co-ordinates with Arg88 and the acyl chain sits in a cleft composed of several hydrophobic residues.

L-arabinose transport through OmpF (set as 100% swelling rate), PopA$_{his}$ was able to act as a pore, but only able to permit the passage of small substrates, notably glycine and alanine (Fig. 3e). These data are in agreement with the lack of a sizable channel in our structures. We were unable to test a FadL-like role for PopA because these assays are performed in-vivo[20] knowing the fatty acid transport profile of the native strain, which is currently obscure for *B. bacteriovorus*. To test PopA

function via a different route, we attempted to delete the *bd0427/popA* gene. There was no observable deficit in *E. coli* predation or lifecycle progression upon *bd0427* deletion from *B. bacteriovorus* HD100 when using host-dependent cultures. We also attempted *bd0427* deletion in a host-independent background[21], but were unable to obtain this despite a reasonable degree of persistence (see methods) – the cells are thus likely using PopA for an OM function, but further

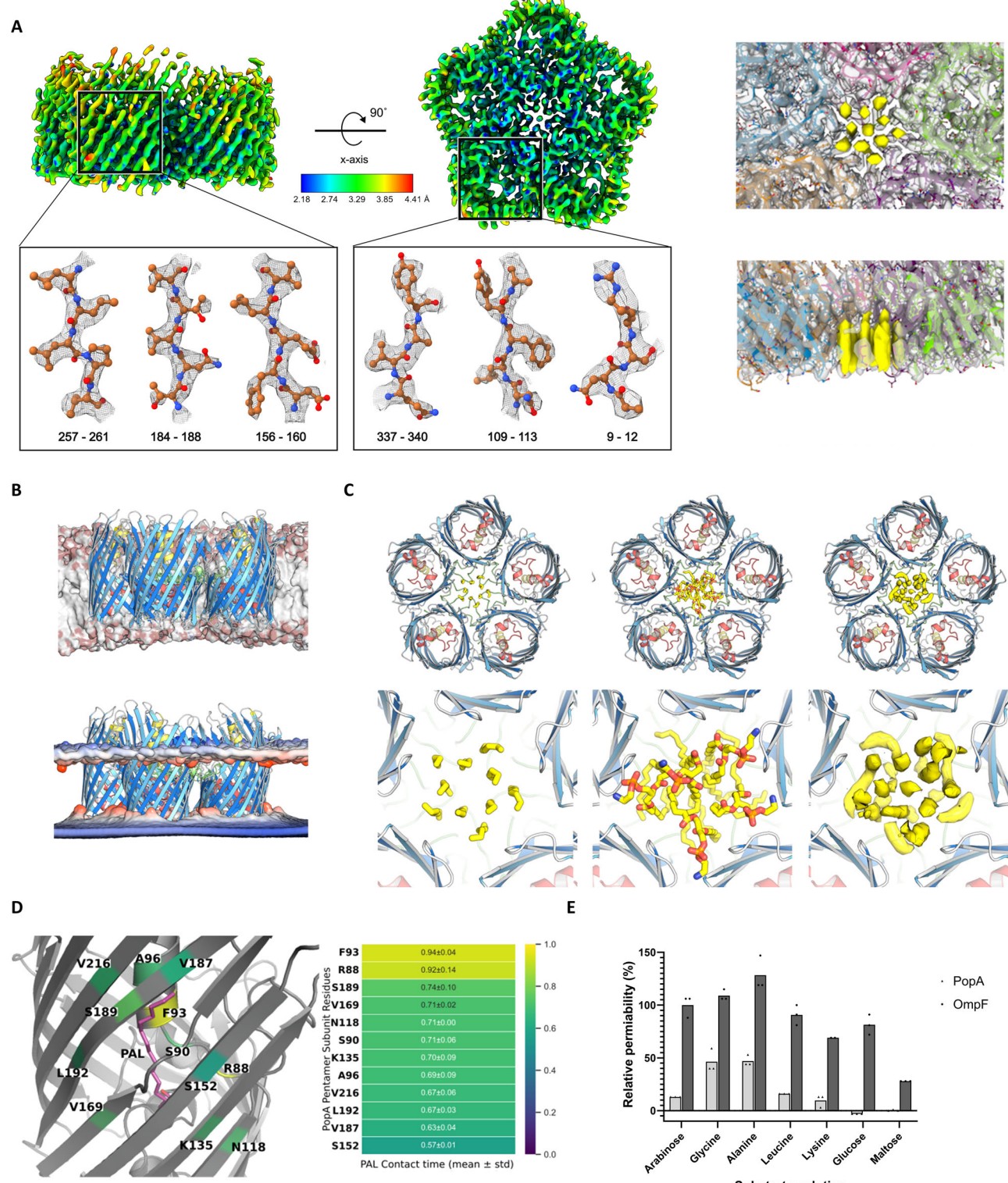

**Fig. 3 | Cryo-EM structure and molecular dynamics of native PopA confirm lipid-trapping features. A** Cryo-EM density map of native PopA$_{his}$, coloured by local resolution. The gross features match those of x-ray determined refolded PopA; the map (yellow, righthand side) is demonstrative of eleven modelled detergent molecules. **B** PopA simulated in a model outer membrane (upper), showing moderate deformation of the bilayer (lower). **C** Comparison of experimental alkyl chains (left), exemplar simulation lipids (centre) and density of lipids averaged over 3 500 ns molecular dynamics simulations (right). **D** Simulation of PopA with palmitate (PAL) and contact time with residues of pocket. **E** Liposome swelling assays of PopA (light bars) relative to *E. coli* OmpF (dark bars), illustrating significant permeability to glycine and alanine only. Liposome swelling assays show an average of three technical replicates.

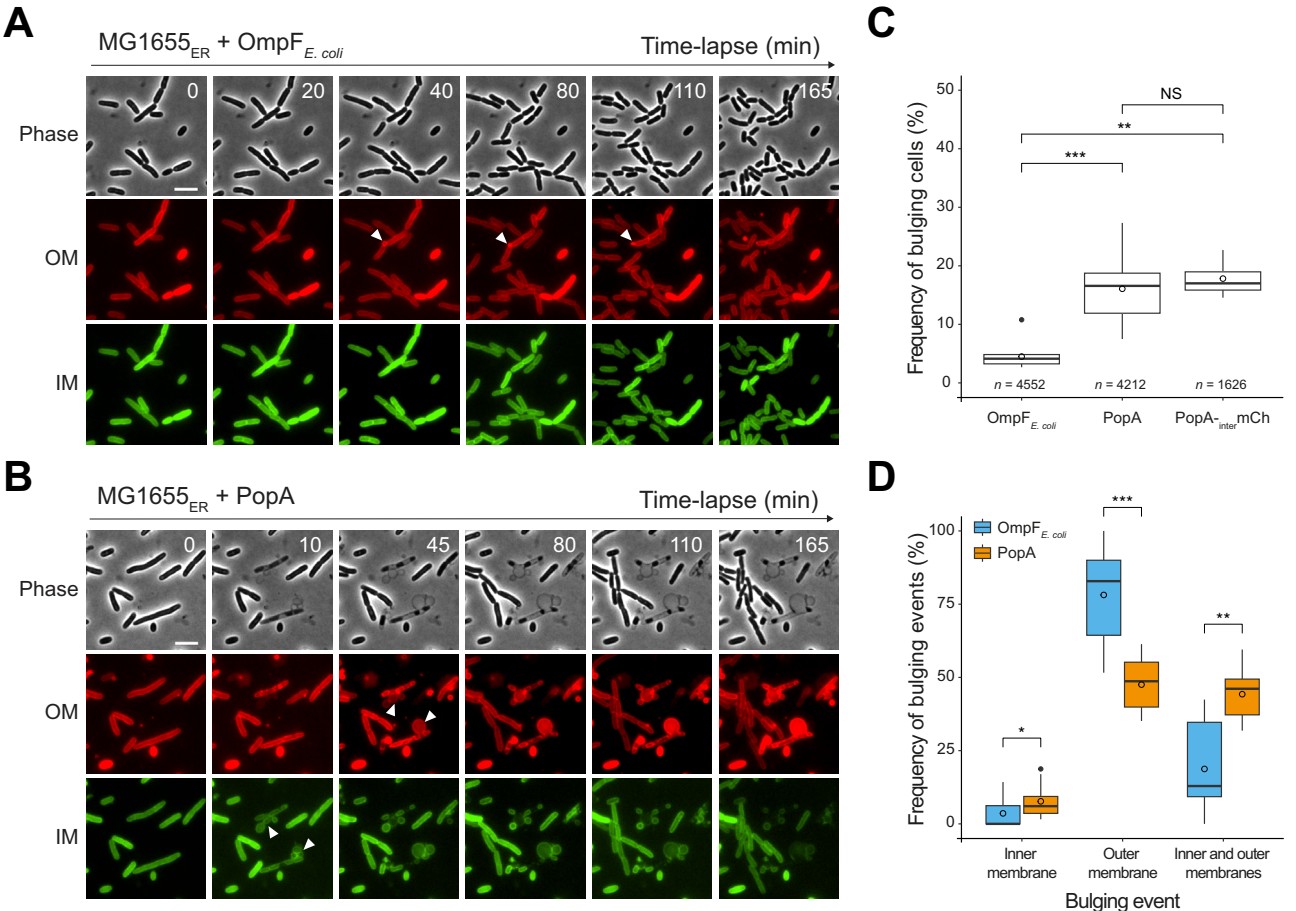

**Fig. 4 | *Bdellovibrio* PopA induces strong envelope defects in *E. coli*.** Representative time-lapse fluorescence microscopy of the *E. coli* MG1655 envelope reporter strain (MG1655$_{ER}$) carrying *lopB-mCherry* (for OM labelling, red) and *msfgfp-glpT* (for IM labelling, green), and producing the OmpF$_{E. coli}$(**A**) or the PopA protein (**B**). Examples of envelope defects (including IM and OM bulging events) are shown (white arrowheads). Note that envelope bulging events were observed at various timepoints throughout the movie. Scale bar, 5 µm. **C** Boxplot representation of the bulging events frequency in *E. coli* cells producing OmpF$_{E. coli}$, PopA or PopA-$_{inter}$mChy. The black dot corresponds to an outlier. The bold horizontal bar represents the median value. Means are represented by the empty circle. The bottom and top boundaries of the internal boxplot correspond to the 25th and 75th percentiles, respectively; whiskers extend 1.5 times the interquartile range

from the 25th and 75th percentiles. Pairwise comparisons from at least three biological replicates are indicated above the plots (NS, nonsignificant; **$p \leq 0.01$; ***$p \leq 0.001$; two-tailed Wilcoxon's t test). OmpF vs PopA: $p = 1.5e{-}6$; OmpF vs PopA-$_{inter}$mChy: $p = 0.0011$; PopA vs PopA -$_{inter}$mChy: $p = 0.52$. The number of analysed cells (*n*) is indicated below for each condition. **D** Boxplot representation of the frequency of IM, OM, or IM and OM bulging events for *E. coli* MG1655$_{ER}$ producing OmpF$_{E. coli}$ (blue) or PopA (orange) protein. Based on the subset of bacteria that exhibited bulging events, as quantified previously (related to Fig. 4C). Pairwise comparisons from at least three biological replicates are indicated above the plots (*$p \leq 0.05$; **$p \leq 0.01$; ***$p \leq 0.001$; two-tailed Wilcoxon's t test). Inner membrane: $p = 0.04374$; Outer membrane: $p = 0.00047$; Inner and outer membrane: $p = 0.00164$.

experimentation is needed to outline a potential role for PopA in this conditional context.

To evaluate the potential impact of PopA in the prey cell envelope, we induced the production of PopA, with its native signal peptide, in *E. coli* MG1655 in which the OM and IM were differentially labelled with fluorescent proteins. Imaging of these cells by phase contrast and fluorescence time-lapse microscopy revealed strong envelope defects, including bulging, unlike those of an *E. coli* OmpF control (Fig. 4a–c). Notably, among the bulging events observed in 16% of PopA-producing cells (Fig. 4b), IM disruptions (or simultaneous IM and OM defects) were significantly more prevalent compared to the few bulging (4.5%) OmpF–producing cells (Fig. 4c), suggestive of a potential interaction of PopA with the *E. coli* IM.

To assess the localization of *B. bacteriovorus* PopA when induced in *E. coli*, we fused the mCherry protein to the same position as the previously used his-tag, resulting in the PopA-$_{inter}$mChy fusion. We first confirmed that the production of PopA-$_{inter}$mChy results in the same percentage of envelope defects as observed with untagged PopA (Fig. 4c). When induced in *E. coli*, PopA-$_{inter}$mChy localizes in distinct

patches, randomly distributed along the cell envelope (Fig. 5); envelope defects were only observed in cells producing PopA (*i.e.*, with a detectable mCherry fluorescent signal). The non-negligible fraction of cells without PopA-$_{inter}$mChy (33.2% ± 2.8, Supplementary Fig. 12) implies that the frequency of PopA-induced bulging calculated in Fig. 4b, c is likely underestimated.

In line with the microscopy data, a sucrose gradient separation of the membrane fraction showed that PopA, when heterologously produced in *E. coli*, localizes in the OM as expected for a β-barrel (PopA, migrating at a size of ~40 kDa) but also in the IM fraction, although with a slower migration (PopA*, ~70 kDa) (Fig. 5). This result is consistent with a native blot analysis of PopA in *B. bacteriovorus* attack phase cells (*i.e.*, in the absence of prey) and during the growth phase inside prey (2 hours post-infection) (Fig. 5).

## Oligomerisation and chamber formation is conserved in PopA relatives and beyond

Because the validated pentamer was distinct from all previously described OMPs, we were interested to compare whether AlphaFold[18]

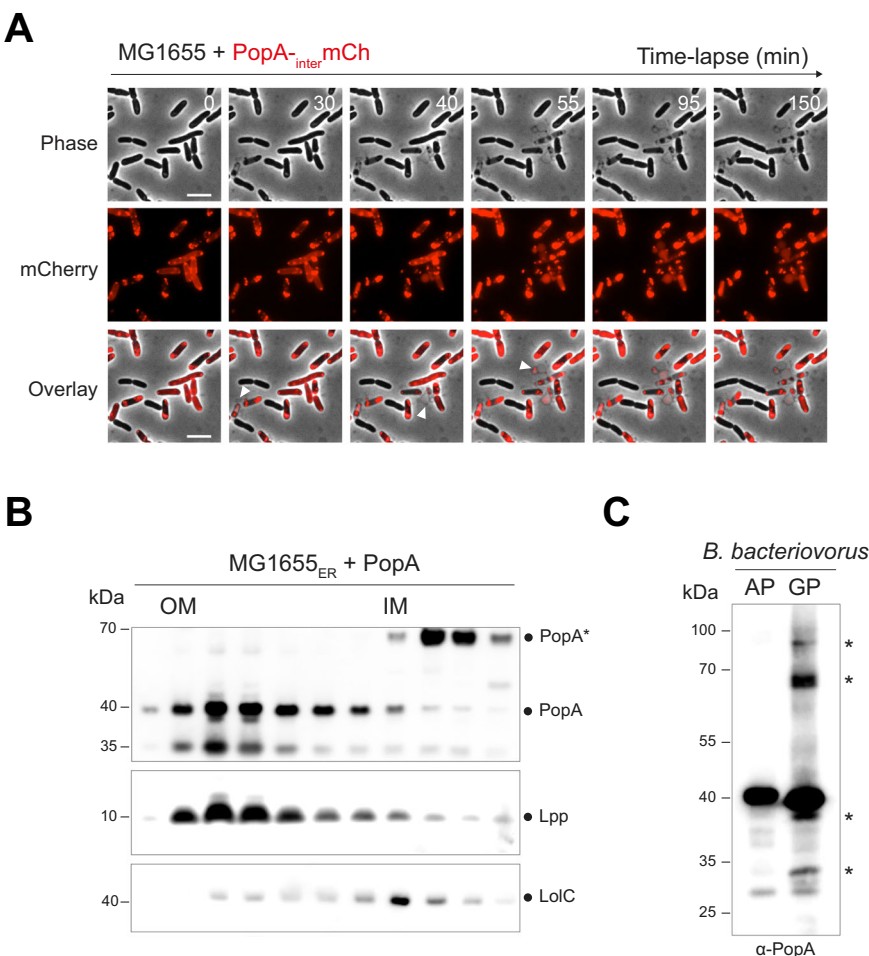

**Fig. 5 | PopA forms discrete patches in the *E. coli* envelope and localizes to both the inner and outer membranes. A** Representative time-lapse fluorescence microscopy of *E. coli* MG1655 producing the PopA-$_{inter}$mChy fusion protein (top panel, phase contrast; middle panel, mCherry channel; bottom panel, overlay of the phase contrast and the mCherry channels). Examples of envelope defects are shown (white arrowheads). Scale bar, 5 μm. Live imaging was performed three times with similar results. **B** The outer and the inner membrane of PopA-producing *E. coli* were separated by centrifugation in a two-step sucrose density gradient (see Methods). PopA was found in two different variants: The PopA protein is primarily localized in the outer membrane of *E. coli* with an expected size of approximately 40 kDa. However, it also exhibits localization in the inner membrane, where it appears as a higher molecular weight form (~65 kDa). Control markers include the

major outer membrane lipoprotein Lpp (rabbit polyclonal antibody, Collet lab) and the inner membrane lipoprotein-releasing system LolC (rabbit polyclonal antibody, Collet lab). Goat anti-rabbit IgG-peroxidase antibody (Sigma) was used as a secondary antibody (0.2 μg/mL final). Molecular weight markers (kDa) are indicated on the left. Fractionation experiments were performed in duplicate with identical results. **C** Western blot of whole-cell protein extracts from *B. bacteriovorus* in attack phase (AP) or in growth phase (GP, 2 hours post-infection) was probed with α-PopA antibodies. The presence of asterisks (*) indicates the different forms of PopA that are exclusively present in the presence of the prey (i.e., in GP). Molecular weight markers (kDa) are indicated on the left. Western blot analysis was independently performed on two biological replicates, yielding consistent results.

would model this in line with our observations – indeed, this occurs with high confidence (Supplementary Fig. 13). PopA was previously identified as the major OMP of strain HD100, with homologues (23-52 % sequence identity) also representing the major OMPs of related strains[11]. These also model confidently as pentamers, including an operon of two putative pentameric OMPs, those of *B. stolpii* (Fig. 6a, differing by 100aa). Beyond predatory strains, FoldSeek[15] identifies homologues in several broad, dispersed bacterial groupings (Supplementary Table 3, and discussion). Intriguingly, some of these homologues expand the scope of the superfamily, modelling confidently as tetramers and hexamers (a selection is shown in Fig. 6a, confidence metrics for all models are presented in Supplementary Fig, 14–21, and models have been deposited at modelarchive.org). Hexamers typically adapt to fill the relatively larger central cavity with longer loops, often including α-helical regions, whereas tetramers pack around a smaller loop 8 (Fig. 6a). Regardless of chamber size, all superfamily members retain the hydrophobicity patterns of lipid interaction observed for PopA.

From manual inspection of membrane proteomes, we were able to find OMPs from a second, vastly-divergent superfamily that models as radial oligomers with a central hydrophobic cavity. These too model confidently as predominantly tetramers-hexamers (Supplementary Table 4, at least one nonamer), and include *Bacteroides thetaiotaomicron* BT1926 (Fig. 6b–e). These are 8-stranded, closed β-barrels (no plug) with no sequence or structure link to the PopA superfamily, that use loop 1 as the central symmetry-forming element. We therefore conclude that oligomerization (via an outer composite β-rim, with seam formation) and subsequent central chamber formation is a general feature found in more than one OMP class.

## Discussion

The pentameric nature of Bd0427/PopA redefines expectations of OMP assembly and interaction with the lipid bilayer. In comparison to trimeric OMPs, PopA does not retain the same ordering of loops and loop function, *i.e.*, trimers use loop 2 for oligomerisation and loop 3 for stabilization of the outer barrel wall, distinct from loop 8

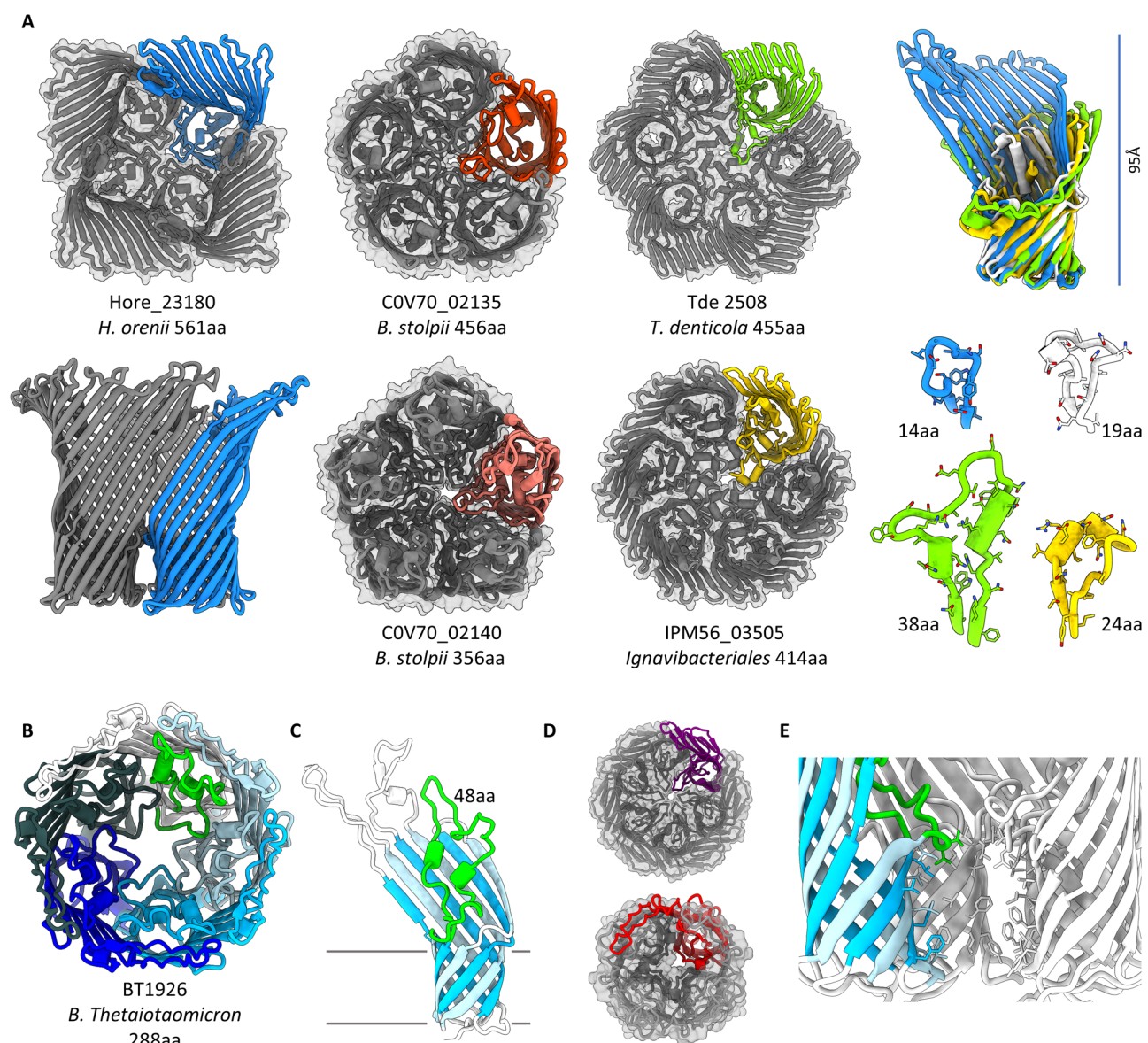

**Fig. 6 | PopA is a representative of wider oligomer/lipid-trapping families.**
**A** Alphafold[18]/Collabfold[62] models for select PopA superfamily members. Gene name, source organism and protein size annotated below. Hexamers from *Fibrobacter*, *Ignavibacter* and *Treponema* demonstrate a larger central chamber than pentameric homologues. Equivalents to the PopA oligomer loop 8 (white, 19aa) are coloured as per monomers, demonstrating that the *Halothermothrix orenii* tetramer (blue) packs around a smaller loop, but hexamers (IPM56_03505, yellow; TDE_2508, green) pack around a larger loop, inclusive of alpha helical regions. The larger size of *H. orenii* Hore_23180 results in a similar barrel, but "taller" outer wall (95 Å high). **B** The smaller, 8-stranded OMP BT1926 models as a radial pentamer (one chain in white, with central loop coloured green). **C** Fold of BT1926 monomer, membrane plane indicated. The large central loop (green) travels "upward" rather than outward. **D** Homologues of BT1926 model as tetramer (red, *D. gadei* HMPREF9455_00941) and hexamer (purple, *C. hutchinsonii* CHU_0007). **E** Cutaway view of central chamber of BT1926, one chain coloured as in C, hydrophobic residues lining chamber shown in stickform.

and loop 2 of PopA, respectively (Fig. 1j, Supplementary Fig. 22). The central chamber of PopA, into which we have modelled detergent alkyl chains, likely houses inner leaflet lipids in vivo. The residues lining the cavity show a high degree of conservation (Supplementary Fig. 4). The density maps from both the x-ray and cryo-EM suggest tight packing of eleven alkyl chains (the central "extra" chain having less defined density in the x-ray map), with chain:chain distances of 4.5 Å commensurate with a crystalline monolayer. Multiple bound lipids are common in several classes of protein, but the ringfence-like trapping of these in PopA is vastly different from the often peripheral groupings of lipids observed in OMPs and closer to the closed-symmetry IM proteins yeast PMA1[22] and the *H. pylori* urea channel[23]. There may be a theme in both IM and OM lipid pools – PMA1 has 115 densities which fit 57 diacylglycerides with 1 random space remaining

to ensure free diffusion[22]; our observation of 11 densities for 5 lipids also matches this pattern.

Because the PopA structure is so distinct, it is interesting to consider oligomer assembly into the OM. The surface bowl-shaped cavity is formed by a continuous β-sheet wall created by the hydrogen bonding between β-barrel strand 5 with strand 14 of the adjacent subunit. Such a structure implies that exclusion of LPS from this space would occur early during PopA subunit assembly by the barrel assembly machine protein A (BamA) and that oligomerisation outweighs the energetic cost of LPS exclusion. Immediately adjacent to L8 is the C-terminal strand of PopA, which contains a β-signal motif that, as shown in canonical OMPs[24], is involved in initiation of folding and membrane insertion upon its hybridisation to the first strand of the BamA β-barrel[25]. The monolayer cavity-forming PopA L8 would

therefore be positioned within the OM early in the subunit assembly process and cause significant membrane disruption. Membrane disruption by L8 could lower the energy cost of LPS expulsion and would be assisted by the membrane-thinning properties previously defined for BamA[26].

The power of structure-based methods to identify distant homologues enabled us to confidently discover other proteins that we propose to form a radial oligomer OMP superfamily distributed throughout bacteria (Supplementary Table 3). Most interestingly, these include prospective tetramers and hexamers as well as pentamers, including some strains that encode multiple types (e.g., *Fibrobacter succinogenes* FSU_2404 hexamer and FSU_0230 pentamer). The size of loop 8 in PopA homologues is a good indicator of oligomerisation state, with tetramer loops typically small (often tight turns) and hexamer loops typically larger with helical content. PopA superfamily members have consistent features: 16 β-strands, a high "outer" wall, a tight plug and loops that create the oligomerisation and bowl effect observed in PopA –making a composite β-sheet wall around the outer rim of the oligomer. Homologues retain the hydrophobicity on loop 8 and strands β12-16 needed to form the central chamber. The ability to use our experimentally-derived PopA structure as a predictive tool is high – even monomeric Alphafold models retain the features described above – *i.e.*, setting the correct oligomerisation state is not a prerequisite to discern PopA-like features. Superfamily members from *Fibrobacter* were observed to be highly expressed during OM vesicle formation (FSU_2404[27]) and cellulose degradation (FSU_0230[28]). Family member TDE_2508 from the human pathogen *Treponema denticola* has been associated with adhesion and biofilm formation[29], and was identified to form high molecular weight oligomers via both chemical crosslinking and blue native PAGE[30] - consistent with modelling as a hexamer (Fig. 6a). One PopA homologue, Hore_23180 (Fig. 6a). occurs in the firmicutes (predominantly Gram-positive), in *Halothermothrix orenii*, which is known to have a Gram-negative diderm phenotype[31]. Additionally, there may even be a distal relative in the Archaea – from DPANN Candidatus *Altiarchaeales* (Supplementary Table 3).

The oligomerization and chamber formation of the PopA superfamily is mirrored by a second, unrelated 8-strand superfamily (Fig. 6b–e, Supplementary Table 4). Representatives have attributed roles in the literature including BT1926 implicated in phase-variable modification of *B. thetaiotaomicron* to alter phage susceptibility[32,33], CHU_0007 implicated in *C. hutchinsonii* adhesion to cellulose[34], and Tanf_04855 found in OM vesicles from the pathogen *Tannerella forsythia*[35]. Remarkably, the thermophilic bacterium *Caldithrix abyssi* has both pentameric and hexameric 16-strand representatives (Cabys_3254 and 2908) and a 8-strand representative (Cabys_3215). It is interesting to note that the two superfamilies together propose a general feature for oligomer formation and lipid binding at the periplasmic chamber. Upon insertion to the OM there will be two consequences – displacing more lipids from the outer leaflet than the inner leaflet may effect physical properties of the OM, but more importantly, cells expressing these OMPs will have a lower density of lipopolysaccharide. This latter outcome may assist processes like adhesion by creating some clearance for other factors to function.

When heterologously produced in *E. coli*, PopA associated with both the OM and the IM, the latter likely mediating the IM blebbing specifically induced by this OMP. IM association may arise during or following secretion to the OM. We favour the latter as it is in keeping with the observed transfer from the *Bdellovibrio* OM during native predation[3,4] and may be an intrinsic property of PopA. It is important to note that under native predation conditions, the proposed transfer of PopA into the prey envelope[3,4] does not result in similar IM defects, hence other factors produced during predation may have stabilizing effects[36,37] or defects could be magnified by PopA overexpression levels. The behaviour of PopA in liposome swelling assays argues that

any role (prior to inter-membrane transfer) in the predator would be unrelated to small-molecule transport. Hence, it may be best to refer to PopA and homologues as porin-like until assays conclusively demonstrate porin behaviour. PopA conformation in the inner membrane of prey has the potential to be influenced by the proton motive force (PMF), but is unlikely to be key for function because the prey PMF has been shown to be dissipated at a very early stage of the predatory lifecycle[38]. We suggest that the reported PopA transfer into prey may occur via the vesiculation observed in tomograms[7]. With PopA possessing FadL-like features and several models of homologues instead having more pore-like features (*e.g.*, TDE_2508 has a clear solvent channel inside the barrel), it is likely that individual oligomeric porins have roles specific to themselves, with the above common physical features of lipid-sequestration and lipopolysaccharide exclusion as an additional functional outcome.

Together, our data reveal that a porin-like protein constituting the major OMP of *B. bacteriovorus* is a representative for a distinct class of membrane proteins. The non-standard features of the pentameric assembly allow for stability in both outer and inner membranes, and the self-enclosed central lipid pool is likely to be a conserved feature of diverse OMP homologues across diverse lifestyles that await further functional characterization.

## Methods

### Cloning and expression

Constructs were generated via restriction-free cloning[39], using primers detailed in Supplementary Table 5. Briefly, PopAΔSP (for refolding) was cloned into pET28a and mature PopA$_{his}$ (for native expression) with an internal His tag (with a *pelB* leader sequence at start and coding sequence SSHHHHHHGSS inserted after A103) was synthesized as an insert into pET29b (Twist Biosciences).

For large-scale expression of PopAΔSP in BL21(DE3) cells, 1 litre of LB media was inoculated with 10 mL overnight culture, incubated at 37 °C until an OD$_{600nm}$ of 0.6, then induced with 1 mM IPTG for a subsequent period of 4 hours. Selenomethionyl-Bd0427ΔSP was produced using the method of Doublie et al. [40].

Native expression of PopA$_{his}$ used BL21(DE3) RIPL cells (New England Biolabs), as per PopAΔSP, except 0.5 mM IPTG and overnight expression were chosen for optimal expression levels.

### Purification of PopAΔSP

Cells were harvested by centrifugation, resuspended in Buffer 1 (50 mM Tris pH 8.0, 5 mM EDTA) and lysed via sonication. The insoluble material was pelleted (43,380 g for 1 h), then resuspended in Buffer 1 plus 2% w/v TritonX-100 for 1 h at room temperature (plus stirring). The pelleting and wash procedures were repeated a further four times (two initial washes with TritonX-100, two subsequent washes without). The isolated pellet of inclusion body material was weighed out and 200 mg tumbled in 10 mL of 50 mM Tris pH 8.0, 100 mM NaCl, 8 M Urea (2 hours at room temperature). Insoluble material was pelleted at 43,380 g for 20 min, protein adjusted to 0.1 mM, and added dropwise to 150 mL refolding buffer (50 mM Tris pH 8.0, 100 mM NaCl and a choice of detergent – either 1% w/v DDM or 2% w/v β-OG). Aggregated protein was removed by filtration and correct refolding assessed by semi-native PAGE. Samples were further purified/analysed via size exclusion chromatography (in varying buffers – the optimal buffer for crystallization sample was 50 mM tris-HCl (pH 8), 150 mM NaCl, 50 mM β-OG, 3 mM β-mercaptoethanol).

### Purification of PopA$_{his}$

Cell pellets were resuspended in 1x PBS, 5 mM EDTA and lysed using an Emulsiflex C3 disruptor (Avastin) at 17,000 psi for 3 cycles. Sequential centrifugation steps were used to isolate cell debris (10,000 g 30 min) and membrane fraction from subsequent supernatant (150,000 g 1 h). Membrane fractions were resuspended in a ratio of 40 mg membrane

to 1 ml buffer (50 mM Tris pH 8.0, 150 mM NaCl, 1% w/v DDM), homogenized by hand homogenizer, and tumbled at 4 °C overnight. Unsolubilized material was pelleted at 150,000 g for 30 min and the supernatant purified by conventional nickel-NTA chromatography using resuspension buffer containing a lower concentration of 0.3% w/v DDM, eluted with 200 mM imidazole. Size exclusion chromatography was performed using a Superdex 200 column.

## PopAΔSP crystallization and structure determination
Purified protein was concentrated to 4.5 mg/ml and screened against a selection of commercial sparse matrix conditions (equal protein:reservoir in drops). Crystals with appreciable diffraction were obtained from MidasPlus C9 (0.1 M NaCl, 25% v/v pentaerythritol propoxylate (5/4 PO/OH), 10% v/v DMSO; form A, C2221 spacegroup), MemGold G12 (1 mM Zinc sulphate, 50 mM HEPES pH 7.8, 28% v/v PEG 600; form B, $P22_12_1$ spacegroup), and MemGold D5 (50 mM Mg acetate, 50 mM Na acetate pH 5.4, 24% PEG 400; form C, C2 spacegroup). Data were collected at the Diamond Light Source synchrotron (beamline IO4), and processed using ISPyB[41] automated software. The CRANK2 pipeline in CCP4 cloud[42] was used to phase and build selenomethionyl form A, which was iteratively refined and built using PHENIX[43] and COOT[44]. Monomer models from form A were used as search models in PHENIX_MR to determine forms B and C via molecular replacement. Data collection and refinement statistics are provided in Supplementary Table 1.

## Cryo-Electron Microscopy of PopA$_{his}$
Quantifoil R1.2/1.3 200-mesh grids were glow discharged for 30 s using a GloQube (Quorum) discharge unit (180 sec at 40 mA) followed by graphene oxide coating. Cryo-EM grids were prepared with a Vitrobot Mark IV (Thermo Fisher) operated at 100% humidity and 4 °C. A 3 µl aliquot of PopA$_{103his}$ (0.02 mg/mL in 50 mM Tris pH 8, 150 mM NaCl, 0.03% w/v DDM) was applied to the grids, blotted 3 s and plunge frozen into liquid ethane.

The samples were imaged using a FEI Titan Krios (300 kV) equipped with a K3 detector and energy filter (20-eV slit size) (Gatan) using automated data collection (ThermoFisher EPU).

Movies were acquired at 105,000x magnification (0.835 Å/pixel, 100 µm objective aperture, 2 s exposure, 50 frames, total dose on specimen -55.3 $e^-/Å^2$ and defocus range of -2.4 to -1.2 in 0.3 µm intervals). A total of 10,308 movies were collected. All data processing was done in cryoSPARC[45] v4.2.1 with workflow displayed in supplementary Fig 7. Raw movies were processed using patch motion correction (multi), with dose weighting. The Contrast Transfer Function (CTF) was estimated by patch CTF estimation (multi). Particles were blob picked, 2D classes were inspected and used as a reference for complete particle picking. A total number of 992,500 particles were subjected to ab initio model generation. Three models were generated in C1 symmetry, one model (365,835 particles) was selected for further processing. The consensus map was refined with non-uniform refinement to a 2.85 Å resolution in C1 symmetry and 2.66 Å in C5 symmetry. Resolution was reported from the gold-standard Fourier shell correlation using the 0.143 criterion. Local resolution was estimated using cryoSPARC[45]. The FSC curve is provided in Supplementary Fig. 7 and local resolution estimation is provided in Supplementary Fig. 8); model-to-data correlation coefficients are provided in Supplementary Table 2.

## Liposome swelling assays
Proteoliposomes were prepared from 80 µl of liposome stock solution (100 mg/ml phosphatidylcholine, 2.3 mg/ml dihexadecyl phosphate in chloroform) dried individually under air flow. The lipid film was left to fully dry under vacuum for 2 hours, then resuspended in 100 µl water before protein addition (15 mg OmpF, 15 mg PopA, 1:1 with detergent buffer (50 mM Tris pH 8.0, 150 mM NaCl, 0.03% w/v DDM)), followed by a vortex step. Samples were sonicated in a water bath for 90 seconds to clarify, then transferred to a 24-well plate to be dried overnight under vacuum. Proteoliposomes were resuspended in 200 µL buffer (10 mM Hepes pH 7.0, 12 mM stachyose) and left to stand for 1–2 hours. Liposome swelling was monitored by following OD$_{400}$ over 30 seconds, with measurements taken every second. Assays contained 15 µl of liposomes/proteoliposomes and 200 µL of substrate solution, with each assay repeated in triplicate. Relative permeability was compared to OmpF in the presence of L-arabinose as a percentage.

## Bacterial strains, growth conditions and chemicals
All strains used in this study are listed in Supplementary Table 5. *Escherichia coli* MG1655 strains were routinely grown in Lysogeny Broth (LB, MP Biomedicals) at 37 °C and constant shaking. Plasmids were maintained by adding ampicillin (100 µg.mL−1) and chloramphenicol (15 µg.mL$^{−1}$).

## Plasmid construction
All primers used in this study are listed in Supplementary Table 6. Standard molecular cloning methods were used, and DNA assembly was performed using the NEBuilder HiFi mix (New England Biolabs). To construct pBAD18 derivative plasmids, the Bd0427-coding sequence from *B. bacteriovorus* HD100 (taxon: 264462) and the OmpF-coding sequence from *E. coli* MG1655 were PCR amplified with primers oGL1727/oGL1728 or oGL1765/oGL1766, respectively, and cloned in the opened pBAD18 vector (using oGL1487/oGL1488) by DNA assembly. The mCherry-coding sequence was then amplified with oGL2261/oGL2262 from the pTNV215-parBBb-mCherry plasmid[46] and cloned in the opened pBAD18-PopA vector (using oGL1488/oGL2256) by DNA assembly.

## ΔpopA strain generation
The scarless allelic replacement of *bd0427/popA* into the HD100 chromosome was performed using a strategy based on the two-step recombination with pK18mobsacB-derived suicide vector, as in Steyert and Pineiro[47], screened by PCR and verified by DNA sequencing. For homology, 500 bp upstream and downstream of the loci of interest were used.

We also screened 10, 26 and 29 colonies from three independent pK18mobsacB:*bd0427*mcherry conjugations into host/prey-independently (HI) grown *B. bacteriovorus* exconjugants again without success. Attempts were also made to construct a C-terminal mCherry fusion to Bd0427 in predatory *B. bacteriovorus* HD100. Although it was possible to construct and verify by sequencing, the fusion plasmid in *E. coli* S17-1, when that was conjugated three times into *B. bacteriovorus* HD100, this resulted in failure to isolate any fluorescently tagged strains despite screening 367, 676 and 346 individual exconjugant plaques. Taken together, these results indicate to us that Bd0427/PopA, is important for viability of *B. bacteriovorus* HD100 in prey-independent growth.

## Image acquisition
Phase contrast and fluorescence images were acquired on a fully motorized Ti2-E inverted epifluorescence microscope (Nikon) equipped with a CFI Plan Apochromat λ DM 100 × 1.45/0.13 mm Ph3 oil objective (Nikon), a Sola SEII FISH illuminator (Lumencor), a Prime95B camera (Photometrics) or a PrimeBSI camera (Photometrics), a temperature-controlled light-protected enclosure (Okolab), and filter-cubes for mCherry and GFP (Nikon). Multi-dimensional image acquisition was controlled by the NIS-Ar software (Nikon). Pixel size was 0.074 µm for the Prime95B camera (using the 1.5X built-in zoom lens of the Ti2-E microscope) or 0.065 µm for the PrimeBSI camera (with 1X zoom lens). Identical LED illumination power and exposure times were applied when imaging several strains and/or conditions in one experiment and across replicates of the same experiment. They were set to the minimum for time-lapse acquisitions to limit phototoxicity.

## Live-cell imaging

To conduct time-lapse imaging, *E. coli* cells were cultured in LB medium supplemented with appropriate antibiotics at 37 °C. In the case of *E. coli* strains producing envelope reporters, 10 μM of IPTG was added to the medium after 30 minutes of initial growth to induce the envelope reporters. Once $OD_{600}$ reached 0.4, cells were incubated with 0.2% L-arabinose for 1 hour to induce the expression of PopA or PopA-$_{inter}$mChy. Subsequently, the cells were harvested at room temperature by centrifugation at 2600 x *g* for 5 minutes, resuspended in LB medium and then imaged on 1.2% DNB-agarose pads supplemented with 10 μM IPTG and 0.2% L-arabinose. The same fields of view on the pad were imaged at 5-minute intervals with the enclosure temperature set to 28 °C.

## Envelope bulging observation and quantification

Envelope bulging was defined as events in which envelope extrusion can be observed. Because envelope bulging can lead to rapid loss of cell integrity and cell lysis, which can happen within the 5-minute interval of our time-lapse experiment, this includes both classical membrane bulging and partial membrane detachment, involving either the inner, outer or both membranes. The frequency of bulging events in each field of view was quantified for each replicate, and then normalized by the total number of cells. In addition, the bulging events were categorized into three categories based on envelope reporters: inner membrane bulging only, outer membrane bulging only, and inner and outer membrane bulging (related to Fig. 4).

## Image processing

For figure preparation, images were processed with FIJI[48] keeping contrast and brightness settings identical for all regions of interest in each figure. Figures were assembled and annotated using Adobe Illustrator (Adobe Inc.).

## Molecular dynamics simulations

All MD simulations were performed using GROMACS v2022[49]. Coarse Grained (CG) MD simulations (martini 2.2[50]) were constructed using Insane[51], using the MemProtMD pipeline[52] and run for 1 μs to permit the equilibration of the OMP within either a symmetrical inner membrane, at a 7:2:1 ratio of 1-palmitoyl-2-oleoyl-phosphatidyl-ethanolamine (POPE): 1-palmitoyl-2-oleoyl-phosphatidyl-glycerol (POPG):Cardiolipin, or an asymmetric outer membrane, comprised of a 7:2:1 POPE:POPG:Cardiolipin leaflet and a pure lipid A core oligosaccharide leaflet[53]. The membrane-assembled systems were then converted to atomic detail[54]. Three independent systems were equilibrated for 10 ns with the protein restrained before repeats of 500 ns of unrestrained atomistic MD simulations using the CHARMM36m force field[55]. Simulations were performed at 310 K, coupled to the V-rescale thermostat[56]. In all cases, the systems were coupled to the stochastic rescaling barostat[57], with the lateral pressure (Pxy) and normal pressure (Pz) coupled independently using semi-isotropic pressure coupling at 1 bar in each dimension. Simulations were analysed using MDAnalysis[58].

## Fractionation assay

Cell fractionation was performed as described herein[59] with some modifications. Cells were grown at 37 °C in LB medium. Once $OD_{600}$ reached 0.4, cells were incubated with 0.2% L-arabinose for 1 hour to induce the expression of PopA. Cells were then collected by centrifugation at 6000 *g* at 4 °C for 15 min, washed and resuspended in PBS buffer. DNase I (1 mg, Roche), RNase A (1 mg, Thermo Scientific) and a half tablet of a protease inhibitor cocktail (Complete EDTA-free Protease Inhibitor Cocktail tablets, Roche) were added to cell suspensions. Cells were broken through a French pressure cell at 1500 psi. After adding 2 mM $MgCl_2$, the lysate was centrifuged at 5000 *g* at 4 °C for 15 min to remove cell debris. Then, clarified supernatant was placed on top of a two-step sucrose gradient (2.3 mL of 2.02 M sucrose in

10 mM HEPES pH 7.5 and 6.6 mL of 0.77 M sucrose in 10 mM HEPES pH 7.5). The samples were centrifuged at 180,000 g for 3 h at 4 °C in a 55.2 Ti Beckman rotor. After centrifugation, the soluble fraction and the membrane fraction (insoluble fraction) were collected. The membrane fraction was diluted four times with 10 mM HEPES pH 7.5. To separate the inner and outer membranes, the diluted membrane fraction was loaded on top of a second sucrose gradient (10.5 mL of 2.02 M sucrose, 12.5 mL of 1.44 M sucrose, 7 mL of 0.77 M sucrose, in 10 mM HEPES pH 7.5). The samples were then centrifuged at 112,000 *g* for 16 h at 10 °C in a SW28.1 Beckman rotor. Approximately 30 fractions of 1.5 mL were collected and odd-numbered fractions were subjected to analysis.

## Native PopA production analysis

To assess the PopA production during the predation cycle, wild-type *B. bacteriovorus* HD100 predators were grown overnight in DNB medium (Dilute Nutrient Broth, Becton, Dickinson, and Company; supplemented with 2 mM $CaCl_2$ and 3 mM $MgCl_2$ salts) in the presence of *E. coli* prey at 30 °C with continuous shaking as described[60]. Attack phase (AP) predator cells were collected from the cleared overnight culture and prepared following the method described by Kaljević[46]. For the growth phase (GP) sample, a 1:1 volume ratio of *E. coli* prey to predator mixture was utilized to minimize the presence of AP cells. Samples from the mixed culture were collected 2 hours post-infection. Western blot analysis of whole-cell protein extracts from *B. bacteriovorus* in AP and in GP was probed with polyclonal α-PopA/ Bd0427 antibodies (produced by Eurogentec) (1/200). Signal for antibody binding was visualized as described in the Immunoblotting section.

## Immunoblotting

Protein samples were separated on 4–12% NuPage Bis-Tris SDS precast polyacrylamide gels (Life Technologies) and ran at 180 V for 1 h in NuPAGE MES SDS running buffer. Western blotting was performed using standard procedures with the following primary antibodies: polyclonal LolC (1/10,000) and Lpp (1/6000) antibodies for LolC and Lpp protein detection, respectively[59], polyclonal Bd0427 antibody (produced by Eurogentec) (1/2000) for Bd0427Bb detection. Signal from antibody binding was visualized by detecting chemiluminescence from the reaction of horseradish peroxidase with luminol and chemiluminescence was imaged with an Image Quant LAS 500 camera (GE Healthcare). Goat anti-rabbit IgG-peroxidase antibody (RABHRP1, Sigma) was used as a secondary antibody. Figures were prepared using ImageJ and assembled and annotated in Adobe Illustrator.

## Statistical analysis

All analyses of microscopy images were performed using several representative fields of view from at least three independent biological replicates. Statistical parameters (means and standard deviations) were calculated in R using the base R functions[61]. Pairwise nonparametric comparisons of datasets were performed using the unpaired Wilcoxon rank-sum test. Significance was defined by $p > 0.05$ (NS), $p \leq 0.05$ (*), $p \leq 0.01$ (**) and $p \leq 0.001$ (***).

## Reporting summary

Further information on research design is available in the Nature Portfolio Reporting Summary linked to this article.

# Data availability

Atomic coordinates have been deposited in with accession codes 9GA1 (x-ray structure), and 9GF0 and EMD-51308 for the cryoEM structure. Molecular dynamics input, output and parameter files are available at https://doi.org/10.5281/zenodo.15696220. Deposited models for each of the homologues are available at ModelArchive (modelarchive.org) with the accession codes ma-n3ds4 (IPM56_03505 hexamer), ma-e64a6 (C0V70_02135 pentamer), ma-uceig (C0V70_02140 pentamer), ma-rkx5z (Tde_2508 hexamer), ma-lgpuq (Hore_23180 tetramer), ma-

c773r (Bt_1926 pentamer), ma-zp266 (HMPREF9455_00941 tetramer) and ma-czwi5 (CHU_0007 hexamer). Source data are provided as Source Data files. Source data are provided with this paper.

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

## Acknowledgements

SGC, ALL RES PR were funded by the Wellcome Trust Investigator Award in Science (209437/Z/17/Z), RP was funded by a Wellcome Trust AAMR studentship. GR was jointly funded by the BBSRC and the University of Birmingham (through the Midlands Integrative Biosciences Training Partnership - Grant No. BB/M01116X/1). TK is supported by BBSRC Research Grant No. BB/S017283/1. PS acknowledges Wellcome (208361/Z/17/Z), MRC (MR/S009213/1), BBSRC (BB/P01948X/1, BB/R002517/1, BB/S003339/1 and BB/Y003187/1), and the Howard Dalton Centre for funding. MD simulations made use of time on ARCHER and JADE granted via the UK High-End Computing Consortium for Biomolecular Simulation, HECBioSim (http://hecbiosim.ac.uk), supported by EPSRC (grant no. EP/R029407/1). PJS acknowledges Sulis at HPC Midlands⁺, which was funded by the EPSRC on grant EP/T022108/1, and the University of Warwick Scientific Computing Research Technology Platform for computational access. MTD is supported by the National Institute of Allergy and Infectious Diseases (R21AI180112), seed funding from the University of Sydney Infectious Diseases Institute and the University of Sydney Drug Discovery Initiative. The research of BvdB was supported by a Wellcome Trust Investigator award (214222/Z/18/Z) and the BBSRC (SPB; BB/R004366/1). YGS was an EMBO post-doctoral fellow and is currently a post-doctoral fellow from the F.R.S.-FNRS. GL is a WELBIO investigator of the WEL Research Institute and a Research Associate of the F.R.S.-FNRS. Work in the Laloux lab is supported by the European Research Council (ERC) under the European Union Horizon 2020 and Horizon Europe research and innovation programmes (ERC StG PREDATOR, Grant agreement No. 802331; ERC CoG VAMPIRE, Grant agreement No. 101171143). We acknowledge the Midlands Regional CryoEM Facility at the Leicester Institute of Structural and Chemical Biology (LISCB), major funding from the MRC (MC_PC_17136). We thank Michaël Deghelt for providing the pTHV037-lpoB-mCherry_msfgfp-glpT, Jessica El Rayes for advice and technical assistance during the fractionation assay, and Jovana Kaljević for providing whole-cell samples of *B. bacteriovorus* proteins. We are grateful to Satya Bhamidimarri for early assistance with the transport assays. We thank Diamond Light Source for beamtime (proposal mx19880) and for facility support during data collection at I04. For the purposes of open access, the corresponding author has applied a CC BY public copyright licence to the Author Accepted Manuscript version arising from this submission.

## Author contributions

A.L.L. and G.L. independently conceived the project and obtained funding (working on the biochemical and in-vivo aspects, respectively). R.J.P. purified, crystallized and solved the structures of the proteins and investigated function (with transport assays undertaken with AS and BvdB) with supervision from A.L.L. and assistance from S.G.C. and L.M. R.E.S. contributed to the manuscript and supervised predator experimentation. PR investigated gene knockout function in the host-independent strains and investigated the feasibility of mCherry gene tagging in the predator. G.R. helped determine the cryo-EM structure of PopA$_{his}$ under the supervision of T.K. M.J. trialled molecular dynamics simulations with (and under supervision from) P.S.; P.S and D.G. performed the MD simulations and analysis. M.T.D. provided insight into the likely folding ramifications of PopA. Y.G.S. investigated PopA expression and properties in-vivo under the supervision of GL.

## Competing interests

The authors declare no competing interests.
