## [Transparent Peer Review file · Nature Communications]

A porin-like protein used by bacterial predators defines a wider lipid-trapping superfamily

Corresponding Author: Professor Andrew Lovering

Version 0:

Reviewer comments:

Reviewer #1

(Remarks to the Author)
NCOMMS-24-79615-T

Summary:

Herein the investigators report the discovery of a new class of pentameric porin-like protein, differing in assembly of known/characterized Gram-negative beta barrels. The investigator use x-ray and cryo-EM structures to view a plugged lumen, and central chamber that forms a vestibule to trap a lipid monolayers. Together with MD simulations and bioinformatic searches, they identify homologues in both predatory and non-predatory species and propose interactions with the membrane. Furthermore recombinant expression of PopA (pentameric outer membrane protein A) in *E. coli* prey leads to IM defects suggesting mechanism of function.

Strengths:

This is a very well-written manuscript that reports on a novel structural fold of bacterial outer membrane protein. It is a very nice, simple structural description of the pentameric porin with comparisons to traditional trimeric porins. The multi-disciplinary use of crystallography, cryoEM, MD simulations and bioinformatics is put together in a digestible manner for people outside the field. The structural presentation is particularly compelling, as is the identification of the superfamilies!

Weaknesses:

There is a lack of understanding what PopA does in the native organism during predation. Or what these "oligomeric monolayer beta barrels" do?

As acknowledged by the authors "To test PopA function via a different route, we attempted to delete the bd0427/popA gene. There was no observable deficit in *E. coli* predation or lifecycle progression upon bd0427 deletion from *B. bacteriovorus* HD100 when using host-dependent cultures. We also attempted bd0427 deletion in a host-independent background, but were unable to obtain this despite a reasonable degree of persistence (see methods) – the cells are thus likely using PopA for an OM function but further experimentation is needed to outline a potential role for PopA in this conditional context." If the authors could more clearly define what the role is -it would greatly enhance this reviewers enthusiasm for the discovery.

The Bv_PopA localizes to the inner membrane and causes bleb formations when expressed in *E.coli* - - the authors could evaluate the expression levels of OmpF and PopA (of course OmpF is native to *E.coli* so there could be trafficking machineries that are limiting in *E.coli* vs *B.vibrio*) - this would help understand the nature of the role of PopA and remove speculation of at artifact of overexpression

The authors comment that this IM localization of PopA in *E.coli* is not observed when it is expressed in *Bd.vibrio* as a predator

Line 363 "It is important to note that under native predation conditions, the proposed transfer of PopA into the prey

envelope^{3,4} does not result in similar IM defects, hence other factors produced during predation may have stabilizing effects. “

Could this be an artifact of overexpression.

follow-up: Line 239 - does the PopA fraction in the IM still contain the signal peptide -that may explain retention and the size abnormality on the SDS-PAGE (Figure 5b).

Suggestion:

line 160

“The protein was successfully 159 solubilized in 1% (w/v) DDM, and was amenable to cryo-EM analysis.” - to give context, it would be better to define what membrane the protein was isolated from rather than having to dig through the methods - this is important for the context of the native folded state.

Line 259 -“The presence of asterisks (*) indicates the different forms 258 of PopA that are exclusively present in the presence of the prey (i.e., in GP)”. what is GP (prey?).

Suggestion to reword some strong text:

1. “We demonstrate that PopA, reported to insert into prey inner membrane, causes defects when directed into Escherichia coli membranes.” -how would PopA be delivered to inner membranes?

2. “...Our work thus defines oligomeric OMP superfamilies, whose deviation from prior structures requires expectation of membrane-interaction motifs and folding models to be revised. “ -very strong statement and it seems that models generated in the study allow for us to “revisit the structures” of previous predicted monomeric OMP models.

3. “...retain the lipid chamber; a similar chamber is formed by an unrelated closed-barrel family, implicating this as a general feature ... “ -this is hard to follow in the abstract as the definition of a closed-barrel vs the PopA pentamer vestibule is hard to envision with the information given in the abstract

Reviewer #2

(Remarks to the Author)

This paper from Parr et al. describes the structure of PopA, an unusual outer-membrane protein in the predatory bacterium *Bdellovibrio bacteriovorus*. PopA is found to form a pentamer, with a monolayer of lipids in the central chamber between protomers. When expressed in *E. coli*, defects are observed in the membranes. Homologs in other species are found, which may form different oligomers such as tetramers and hexamers.

The structure is indeed quite interesting, but I found the functional claims to be unsupported. More work is needed to improve the impact of the paper overall.

1) For example, it's found in liposome swelling assays that the permeability of PopA is low relative to OmpF. However, I could imagine that in vivo, there is a conformational change that makes it more permeable. Nothing is known about how it gets inserted into the prey's inner membrane; this process, or even the distinct prey environment (perhaps a membrane potential?), could cause such a change. Indeed, it's noted in the discussion that transfer to the prey envelope doesn't induce the same IM defects seen in the *E. coli* expression experiments, bringing the value of these into question.

To be more convincing, I think experiments probing the stability of the plug domain in each monomer as well as the loops in the center in a native context are necessary. The authors do hedge in the discussion by calling it “porin-like” for the time being, but this just emphasizes the lack of clear functional data.

2) Why is there a notable IM fraction of PopA in the *E. coli* expression experiments? Chaperones should prevent OMPs from getting to the IM. Is it that PopA is not a native *E. coli* protein that causes it to be trafficked to the OM less efficiently?

3) Very little analysis from the simulations is shown. For example, it's stated that the monomer simulations displays more fluctuations for the extracellular loops, but this isn't seen anywhere. Additionally, I'm left wondering how stable the central loops are, as well as the pentameric structure overall. In general, I would appreciate more analysis of these.

Reviewer #3

(Remarks to the Author)

Overall summary:

In this well-written manuscript, Parr and colleagues have presented a novel architecture of a porin-like protein, PopA (Bd0427) found in *Bdellovibrio bacteriovorus*. The pentameric organisation of Bd0427 is unique and remarkable, bearing “lipid-trapping” features and a central chamber that intriguingly has a thickness similar to that of a lipid monolayer. In valiant efforts to characterise the mechanisms and function of PopA, Parr and colleagues utilised liposome swelling transport assays and demonstrated remarkable selectivity of PopA in the passage of Gly/Ala, but not sugars unlike OmpF. It was interesting to note that deletion of bd0427 had no implications on predation or viability of *B. bacteriovorus*, possibly hinting that PopA could have a more downstream role in the predation cycle. Coherently, confocal studies pointed at strong envelope defects and swelling in prey owing to PopA induction. Localisation studies serendipitously revealed that PopA is situated at both the OM and IM in prey, possibly in two distinct states that bears different migration rates. Nonetheless, Parr and colleagues objectively proposed that PopA may aid in processes like membrane adhesion and attachment via outer-leaflet lipid displacement and LPS dispersal. Finally, from the unique architecture of PopA, Parr and colleagues have intriguingly identified two distinct novel classes of higher oligomeric porins; a class of 16 β -strand PopA homologues and 8 β -strand BT1926 superfamily across dispersed bacterial groups that were rigorously and confidently modelled as tetramers, pentamers, and hexamers. These superfamilies were likewise proposed to have FadL- and OmpT-like roles. These findings present an interesting avenue for future investigation.

This manuscript represents a significant advance from preliminary studies first reported in Tudor et al., 1994, Beck et al., 2004, and Beck, 2005, and provides molecular/mechanistic details involving prey membrane dynamics complementary to Barel et al., 2005, and cryoET studies in M Kaplan et al., 2023. No major issues towards publication, but a few minor comments and questions below.

General questions/comments:

Line 169: Trimeric porins have stable interactions with annular LPS, especially at positively-charged sites (Arunmanee et al. 2016). It would be interesting if, this interaction is conserved for PopA, and LPS is able to modulate the conformation (e.g. extracellular loops) and function of PopA. Speculatively, it would also be interesting if host-dependent Bd LPS is able to modulate the activity of PopA (i.e. cause swelling faster/alter permeability) over host-independent LPS in the context of prey resource depletion (Schwudke et al., 2003). Nevertheless, would there be a possibility of any stable LPS interaction with PopA, especially at Lys residues described in Fig 1E? Can this be observed in MD simulations?

Line 191-192: Perhaps for a better flow, consider shifting Figure 2C to end of Figure 3 or start of Figure 4. Could the authors also comment on the implications of substrate passage selectivity towards Gly/Ala? Could this have an influence on swelling observed in Fig 4 (perhaps osmolarity)?

Line 226: Should be “Figure 2C”?

Line 238-239: In Fig 5B, Parr et al. show that PopA localised to both the IM (70 kDa) and OM (40 kDa) have different migration rates even in the presence of SDS, perhaps due to differences in conformation at the IM (slower migration) and/or possible annular LPS-binding to PopA at the OM (faster migration). These effects could be preserved even after purification with DDM (as with Fig 5B in presence of SDS). Parr et al. explains that IM-blebbing caused by this IM variant (70 kDa) is an artefact due to the lack of stabilising factors present during predation (line 366). However, this 70 kDa IM-variant is also observed in Fig 5C alongside other GP-present variants, during growth phase of Bd in host-dependent culture, and is likely still physiologically relevant to predation. Hence, it may be interesting to pursue if the IM variant (70 kDa) is conformationally/functionally different from the OM variant (40 kDa).

Following which my questions are: Firstly, are there two protein peaks observable on SEC during the purification of PopA(his)? If only a single chromatographic peak is observed (similar hydrodynamic volumes of IM- and OM-variants), would it be viable to revisit the cryoEM pipeline to isolate volumes corresponding to the second variant (e.g. revisiting discarded particles, or 3D variability analysis / 3D classification of the deposited C1 volume)?

The authors have made commendable efforts to account for this serendipitous observation in Fig. 5C. Do consider including these predation experiment details in the Materials and Methods.

Line 270: In Supplementary Table S3 and S4, it may be clearer to include the length of chamber loop next to size of OMP under the same column (i.e. [Length of chamber loop/Size]) to emphasise ratios that support the direct correlation of chamber loop lengths to larger stable oligomer states. Also, it may be helpful to include PISA values together with ipTM/pTM values, to indicate maximum barrel-barrel SA contact in the optimal configuration (for top few tetrameric/ pentameric/ hexameric hits).

Line 370: Parr et al. propose that PopA could be transferred from predator to prey via vesicles of ~25 nm in prey periplasm (M Kaplan et al., 2023). In this manuscript, the role of PopA in the Bd predation cycle has been proposed to be involved in lipid-trapping and LPS-dispersion, likely causing prey OM-swelling, yet popA deletion in Bd has no impact on predation (line 196-197). Is it likely that PopA has a role in the end-stage lysis of prey bacteria and predator release by contributing to membrane pliability?

Line 649: For EM structure deposition, PDB code provided without EMD code.

Version 1:

Reviewer comments:

Reviewer #1

(Remarks to the Author)

The Authors have addressed my comments - I have nothing further to add.

Reviewer #2

(Remarks to the Author)

I appreciate the authors' hard work on this challenging protein. Although there are still many questions (to be addressed in the coming years!), I feel the added data and analysis have improved the paper a lot and am now happy to support publication.

Reviewer #3

(Remarks to the Author)

In this revision, Parr and colleagues have improved their manuscript by supplementing new data, and have extensively addressed reviewers' comments and incorporated said recommendations. This study would be a great addition to inspire future work towards further characterisation of these novel OMP superfamilies and detailed function-mechanism of Bd0427 in predatory *Bdellovibrio bacteriovorus*, towards innovative biotherapeutics. Please find my unreserved support for the publication of this manuscript.

Line 185: "demonstrating high contact residency time for this ligand during simulations (Fig. 3e)" - should be Fig. 3d?

#We thank Nature Communications staff and anonymous reviewers for their appraisal and feedback. Our responses to individual points are prefaced with ## below, the reviewer comments bracketed by quotation marks.

Reviewer #1 (Remarks to the Author):

NCOMMS-24-79615-T

“Herein the investigators report the discovery of a new class of pentameric porin-like protein, differing in assembly of known/characterized Gram-negative beta barrels. The investigator use x-ray and cryo-EM structures to view a plugged lumen, and central chamber that forms a vestibule to trap a lipid monolayers. Together with MD simulations and bioinformatic searches, they identify homologues in both predatory and non-predatory species and propose interactions with the membrane. Furthermore recombinant expression of PopA (pentameric outer membrane protein A) in E. coli prey leads to IM defects suggesting mechanism of function.

Strengths:

This is a very well-written manuscript that reports on a novel structural fold of bacterial outer membrane protein. It is a very nice, simple structural description of the pentameric porin with comparisons to traditional trimeric porins. The multi-disciplinary use of crystallography, cryoEM, MD simulations and bioinformatics is put together in a digestible manner for people outside the field. The structural presentation is particularly compelling, as is the identification of the superfamilies!”

##We thank the reviewer for this enthusiasm of our characterization and communication of the novel superfamilies.

Weaknesses:

There is a lack of understanding what PopA does in the native organism during predation. Or what these “oligomeric monolayer beta barrels” do? “

##This is a fair question, but with a likely complex answer. As noted in lines 394-5, we think that different homologues in different organisms will have differing roles (rather than a sole role), but all commonly utilising the unprecedented oligomerisation concept to effect the outer membrane protein:LPS exposure ratio, as a physical function/phenotype.

“As acknowledged by the authors “To test PopA function via a different route, we attempted to delete the bd0427/popA gene. There was no observable deficit in E. coli predation or lifecycle progression upon bd0427 deletion from B. bacteriovorus HD100 when using host-dependent cultures. We also attempted bd0427 deletion in a host-independent background, but were unable to obtain this despite a reasonable degree of persistence (see methods) – the cells are thus likely using PopA for an OM function but further experimentation is needed to outline a potential role for PopA in this conditional context.” If the authors could more clearly define what the role is -it would greatly enhance this reviewers enthusiasm for the discovery.”

##We thank reviewer 1 for noting a reasonable degree of persistence in our efforts here. We tentatively suggest a putative FadL function (line 182-185), as supported by both homology

to known FadL folds (lines 89-92, 123) and observation of lipidic density on the inner face of PopA (figure 2h). We now add to this with new data that demonstrate a high residency time for FadL fatty acid ligands at the pocket containing the density (new fig 3d). Prior to our study, there were two publications on the Bd0427/PopA membrane transfer phenomenon (refs 3&4 in our main text), and neither were able to ascertain function. We went further by investigating gene deletion in two different growth modes of *Bdellovibrio*, HD host-dependent growth and HI host-independent growth, our reasoning being that some knockout phenotypes manifest only in one setting (e.g. PMID:26626559 for HD and PMID:22319440 for HI). Despite these efforts, we were unable to confirm function via this route. A further route of probing function through overexpression is not possible “in self” because PopA is one of the most populous proteins in strain HD100 (PMID:15668021), so we chose to take advantage of the predation lifestyle of *Bdellovibrio* and thus monitor overexpression in prey. Our observations (in Fig 4) suggest PopA causes statistically significant membrane defects. For a study centred on description of a new membrane protein superfamily, we believe that although the above falls short of absolute mechanistic function, the FadL similarity (fold, ligand density, MD simulation) and membrane-defect from expression (Figs 4&5) are sufficiently descriptive of PopA characterization.

“The *Bv_PopA* localizes to the inner membrane and causes bleb formations when expressed in *E.coli* - - the authors could evaluate the expression levels of OmpF and PopA (of course OmpF is native to *E.coli* so there could be trafficking machineries that are limiting in *E.coli* vs *B.vibrio*) - this would help understand the nature of the role of PopA and remove speculation of an artifact of overexpression”

##As noted above, the overexpression in prey route is the main option to elicit a phenotype, and tallies with the references below that found PopA in prey during regular predation. While we did not evaluate relative OmpF and PopA protein levels directly, since as the reviewer points out, distinct routes may be involved even if protein levels were similar, we note that ompF could be overexpressed in *E. coli* with no reported cellular defects (see, e.g., Ghosh et al PMID: 9840808).

“The authors comment that this IM localization of PopA in *E.coli* is not observed when it is expressed in *Bd.vibrio* as a predator”

##The reviewer is slightly mistaken here, PopA indeed localizes to IM during predation (PMID:8106336, also PMID:15601717), inclusive of the choice of *E. coli* as the particular prey strain used. Hence us trying the expression in prey route above.

“Line 363 “It is important to note that under native predation conditions, the proposed transfer of PopA into the prey envelope does not result in similar IM defects, hence other factors produced during predation may have stabilizing effects. “

Could this be an artifact of overexpression.”

##We apologise for brevity here and have included two new references that refer the reader (and reviewers) to some discussion about potentially stabilizing effects (line 384). One of these concerns peptidoglycan modifications demonstrated to toughen the invaded cell (PMID:28974693) and the other concerns general envelope modifications known to occur early on in predation (PMID:6202674). We recognize that the defects may still be partially a result of overexpression so have amended the text to raise this as a possibility: line 385

“hence other factors produced during predation may have stabilizing effects, or defects could be magnified by PopA overexpression levels”.

“follow-up: Line 239 - does the PopA fraction in the IM still contain the signal peptide -that may explain retention and the size abnormality on the SDS-PAGE (Figure 5b).”

##We thank the reviewer for this suggestion but consider it unlikely because the proteomic coverage of PopA in prey found no peptides from the signal peptide region (PMID:15090518), consistent with it having been processed; secondly the construct used by us utilizes the PelB signal sequence, native to *E. coli* and was fully processed as demonstrated in our cryoEM purifications/structure of PopA_{his}. More importantly, the first residue of mature PopA, S21 in the signal peptide sequence -AMA_S²¹KA- is buried and it's processed N-terminus makes a strong ionic interaction with the conserved D35 COO⁻ sidechain, strongly suggesting that unprocessed PopA would aggregate/misfold and not be present in the membrane. To better communicate the burial and multiple contacts of this region, we include a figure of this as new Supplementary Figure 5.

“Suggestion: line 160

“The protein was successfully solubilized in 1% (w/v) DDM, and was amenable to cryo-EM analysis.” - to give context, it would be better to define what membrane the protein was isolated from rather than having to dig through the methods - this is important for the context of the native folded state.”

##We do not isolate individual membranes during purification - the extracting detergent is added to a centrifugation pellet of all isolated membranes. Given the 0.7-1.1 Å RMSD between the x-ray and the cryoEM forms, the similarity of both to AlphaFold models (supplementary fig 9), and the identical behaviour of PopA in membrane simulations independent of bilayer type (new supplementary figures 9-11), we do not think that any native folded state will vary significantly from these.

“Line 259 -“The presence of asterisks (*) indicates the different forms 258 of PopA that are exclusively present in the presence of the prey (i.e., in GP)”. what is GP (prey?).”

##Apologies, this is nomenclature from the Bdellovibrio field that we should have explained more clearly. This has now been amended (line 269) to read “or in growth phase (GP, 2 hours post-infection)”, which describes the period of growth following invasion and before septation.

“Suggestion to reword some strong text:

1. “We demonstrate that PopA, reported to insert into prey inner membrane, causes defects when directed into Escherichia coli membranes.” -how would PopA be delivered to inner membranes?”

##The above statement is worded carefully to implicate BOTH inner and outer membranes as observed in figure 5. Expression in *E. coli* (heterologously) results in a population of PopA in inner membranes, mirroring that observed during predation (natively). Hence delivery must be intrinsic; one possibility would be vesicular (noted at line 392). To add this possibility carefully, following the existing line 378 “When heterologously produced in *E. coli*, PopA associated with both the OM and the IM, the latter likely mediating the IM blebbing specifically induced by this OMP” we have now added “IM association may arise during or following secretion to the OM, we favour the latter as it is in keeping with the observed

transfer from the Bdellovibrio OM during native predation and may be an intrinsic property of PopA”.

“2. “...Our work thus defines oligomeric OMP superfamilies, whose deviation from prior structures requires expectation of membrane-interaction motifs and folding models to be revised. “ -very strong statement and it seems that models generated in the study allow for us to "revisit the structures" of previous predicted monomeric OMP models.”

##We agree and have reworded this to “Our work thus defines oligomeric OMP superfamilies, whose deviation from prior structures requires us to revisit existing membrane-interaction motifs and folding models”.

“3. “...retain the lipid chamber; a similar chamber is formed by an unrelated closed-barrel family, implicating this as a general feature ... “ -this is hard to follow in the abstract as the definition of a closed-barrel vs the PopA pentamer vestibule is hard to envision with the information given in the abstract”

##This has been reworded to “a similar chamber is formed by an unrelated smaller barrel family, implicating this as a general feature”.

Reviewer #2 (Remarks to the Author):

“This paper from Parr et al. describes the structure of PopA, an unusual outer-membrane protein in the predatory bacterium Bdellovibrio bacteriovorus. PopA is found to form a pentamer, with a monolayer of lipids in the central chamber between protomers. When expressed in E. coli, defects are observed in the membranes. Homologs in other species are found, which may form different oligomers such as tetramers and hexamers.

The structure is indeed quite interesting, but I found the functional claims to be unsupported. More work is needed to improve the impact of the paper overall.”

##We thank the reviewer for their interest in the structure and respond to additional points below.

“1) For example, it's found in liposome swelling assays that the permeability of PopA is low relative to OmpF. However, I could imagine that in vivo, there is a conformational change that makes it more permeable. Nothing is known about how it gets inserted into the prey's inner membrane; this process, or even the distinct prey environment (perhaps a membrane potential?), could cause such a change. Indeed, it's noted in the discussion that transfer to the prey envelope doesn't induce the same IM defects seen in the E. coli expression experiments, bringing the value of these into question.

To be more convincing, I think experiments probing the stability of the plug domain in each monomer as well as the loops in the center in a native context are necessary. The authors do hedge in the discussion by calling it "porin-like" for the time being, but this just emphasizes the lack of clear functional data.”

##When solving the initial x-ray structure, we too wondered about the possibility that the central loops might move. However, in three different crystal forms (lower resolution, line 74,75) these retain an identical conformation, and we see these reproduced in the cryoEM,

simulations and AlphaFold model; the same is also true of the plug, which is given additional support via its homology to the plug/hatch of FadL which also doesn't move/adopt a different conformation (PMID:19182779). Furthermore, the large number of contacts for both the central loops and the plug, and low relative B-factors suggest a single, stable conformation. The pentamer-induced stability of the loops is demonstrated in new supplementary figures 9-11.

We thank reviewer 2 for raising the issue of membrane potential – this is a very pertinent point, which we had thought partly about, and have now included this possibility at line 388-391, “PopA conformation in the inner membrane of prey has the potential to be influenced by the proton motive force (PMF), but is unlikely to be key because the prey PMF has been shown to be dissipated at a very early stage of the predatory lifecycle (PMID:4908670).

“2) Why is there a notable IM fraction of PopA in the E. coli expression experiments? Chaperones should prevent OMPs from getting to the IM. Is it that PopA is not a native E. coli protein that causes it to be trafficked to the OM less efficiently?”

##This is an intrinsic property of PopA, as outlined in response to reviewer 1. PopA is in the IM, in keeping with observations of PopA transfer during predation (PMID:8106336, PMID:15601717). The chaperone suggestion is another line of support towards the hypothesis that OM insertion and folding precedes (eventual) transfer to the IM.

“3) Very little analysis from the simulations is shown. For example, it's stated that the monomer simulations displays more fluctuations for the extracellular loops, but this isn't seen anywhere. Additionally, I'm left wondering how stable the central loops are, as well as the pentameric structure overall. In general, I would appreciate more analysis of these.”

##Apologies for not discussing this more at length in the original draft. We now include an extra three supplementary figures describing the simulations (new Supplemental Figures 9, 10, 11), including monomer fluctuations that strongly support a model for pentamer-induced stability. As noted above, the central loops are stable in the pentamer assembly, and have even greater stability (buried contact area) in the hexameric counterparts of Fig 6.

Reviewer #3 (Remarks to the Author):

“Overall summary:

In this well-written manuscript, Parr and colleagues have presented a novel architecture of a porin-like protein, PopA (Bd0427) found in *Bdellovibrio bacteriovorus*. The pentameric organisation of Bd0427 is unique and remarkable, bearing “lipid-trapping” features and a central chamber that intriguingly has a thickness similar to that of a lipid monolayer.”

##We thank reviewer 3 for noting the remarkable nature of this family. In the time since submission, we have been able to find even more members and have updated supplementary tables S3 & S4 to reflect this. Excitingly, these include a protein from the human pathogen *Capnocytophaga*, and strains that include a representative from both the 16-stranded and 8-stranded families (text added at lines 370-2 in Discussion to highlight this finding).

“In valiant efforts to characterise the mechanisms and function of PopA, Parr and colleagues utilised liposome swelling transport assays and demonstrated remarkable selectivity of PopA_{his} in the passage of Gly/Ala, but not sugars unlike OmpF. It was interesting to note

that deletion of bd0427 had no implications on predation or viability of *B. bacteriovorus*, possibly hinting that PopA could have a more downstream role in the predation cycle.”

#We thank the reviewer for their appreciation of our characterization, especially in the context of challenging predation phenotyping.

“Coherently, confocal studies pointed at strong envelope defects and swelling in prey owing to PopA induction. Localisation studies serendipitously revealed that PopA is situated at both the OM and IM in prey, possibly in two distinct states that bears different migration rates. Nonetheless, Parr and colleagues objectively proposed that PopA may aid in processes like membrane adhesion and attachment via outer-leaflet lipid displacement and LPS dispersal. Finally, from the unique architecture of PopA, Parr and colleagues have intriguingly identified two distinct novel classes of higher oligomeric porins; a class of 16 β -strand PopA homologues and 8 β -strand BT1926 superfamily across dispersed bacterial groups that were rigorously and confidently modelled as tetramers, pentamers, and hexamers. These superfamilies were likewise proposed to have FadL- and OmpT-like roles. These findings present an interesting avenue for future investigation.”

##We thank reviewer 3 for their enthusiasm and refer to the above statement extending the scope of homologues even further. We are hopeful that our identification and presentation of these inspire other labs to study them.

“This manuscript represents a significant advance from preliminary studies first reported in Tudor et al., 1994, Beck et al., 2004, and Beck, 2005, and provides molecular/mechanistic details involving prey membrane dynamics complementary to Barel et al., 2005, and cryoET studies in M Kaplan et al., 2023. No major issues towards publication, but a few minor comments and questions below.”

##Again, we are grateful for the reviewer’s familiarity and contextualisation of the literature/history of predation – the cryoET work by Kaplan (via its observations of prominent & frequent vesicular events) may even hint at route of PopA transfer.

“General questions/comments:

Line 169: Trimeric porins have stable interactions with annular LPS, especially at positively-charged sites (Arunmanee et al. 2016). It would be interesting if, this interaction is conserved for PopA, and LPS is able to modulate the conformation (e.g. extracellular loops) and function of PopA. Speculatively, it would also be interesting if host-dependent Bd LPS is able to modulate the activity of PopA (i.e. cause swelling faster/alter permeability) over host-independent LPS in the context of prey resource depletion (Schwudke et al., 2003). Nevertheless, would there be a possibility of any stable LPS interaction with PopA, especially at Lys residues described in Fig 1E? Can this be observed in MD simulations?”

##We suspect that the Lys ring (like that of the only other membrane-transfer porin characterised to date PorB) is a general lipid headgroup binder – we can confirm we see phosphate contacts in simulations with *E. coli* LPS but *Bdellovibrio* LPS replaces the consensus phosphate groups with mannose (PMID:12743115), making it neutral. We tried binding assays between PopA and mannose, but did not observe a significant interaction.

“Line 191-192: Perhaps for a better flow, consider shifting Figure 2C to end of Figure 3 or start of Figure 4.”

##This is an excellent suggestion - we have moved Figure 2C to the end of figure 3, and agree it improves the flow of the manuscript, thank you.

“Could the authors also comment on the implications of substrate passage selectivity towards Gly/Ala? Could this have an influence on swelling observed in Fig 4 (perhaps osmolarity)?”

##We think this selectivity is purely a size effect; some of the swelling in Fig 4 could result from this, yes.

“Line 226: Should be “Figure 2C”?”

##We apologise for this error – it should read “Figure 4C” and we have now amended this.

“Line 238-239: In Fig 5B, Parr et al. show that PopA localised to both the IM (70 kDa) and OM (40 kDa) have different migration rates even in the presence of SDS, perhaps due to differences in conformation at the IM (slower migration) and/or possible annular LPS-binding to PopA at the OM (faster migration). These effects could be preserved even after purification with DDM (as with Fig 5B in presence of SDS).”

##We thank reviewer 3 for this suggestion for the migration interpretation.

“Parr et al. explains that IM-blebbing caused by this IM variant (70 kDa) is an artefact due to the lack of stabilising factors present during predation (line 366). However, this 70 kDa IM-variant is also observed in Fig 5C alongside other GP-present variants, during growth phase of Bd in host-dependent culture, and is likely still physiologically relevant to predation. Hence, it may be interesting to pursue if the IM variant (70 kDa) is conformationally/functionally different from the OM variant (40 kDa).

Following which my questions are: Firstly, are there two protein peaks observable on SEC during the purification of PopA(his)? If only a single chromatographic peak is observed (similar hydrodynamic volumes of IM- and OM-variants), would it be viable to revisit the cryoEM pipeline to isolate volumes corresponding to the second variant (e.g. revisiting discarded particles, or 3D variability analysis / 3D classification of the deposited C1 volume)?”

##This is a good point but we see no alternative peaks in the SEC and no alternative states in the cryoEM, just a slight breathing motion.

“The authors have made commendable efforts to account for this serendipitous observation in Fig. 5C. Do consider including these predation experiment details in the Materials and Methods.”

##We thank reviewer 3 for this view on our effort herein, and have extended the methods at lines 603-614 to document this:

“Native PopA production analysis

To assess the PopA production during the predation cycle, wild-type *Bdellovibrio bacteriovorus* HD100 predators were grown overnight in DNB medium (Dilute Nutrient Broth, Becton, Dickinson, and Company; supplemented with 2 mM CaCl₂ and 3 mM MgCl₂ salts) in the presence of *E. coli* prey at 30°C with continuous shaking as described (Remy, O., Lamot, T., Santin, Y., Kaljević, J., de Pierpont, C., and Laloux, G. (2022). An optimized workflow to measure bacterial predation in microplates. STAR Protoc 3, 101104. <https://doi.org/10.1016/j.xpro.2021.101104>). Attack phase (AP) predator cells were collected from the cleared overnight culture and prepared following the method described in (Kaljević,

J., Saaki, T.N.V., Govers, S.K., Remy, O., van Raaphorst, R., Lamot, T., and Laloux, G. (2021). Chromosome choreography during the non-binary cell cycle of a predatory bacterium. *Current Biology*, 1–14. <https://doi.org/10.1016/j.cub.2021.06.024>. For the growth phase (GP) sample, a 1:1 volume ratio of *E. coli* prey to predator mixture was utilized to minimize the presence of AP cells. Samples from the mixed culture were collected 2 hours post-infection. Western blot analysis of whole-cell protein extracts from *B. bacteriovorus* in AP and in GP was probed with polyclonal α -PopA antibodies (1/200). Signal for antibody binding was visualized as described in the Immunoblotting section.”

“Line 270: In Supplementary Table S3 and S4, it may be clearer to include the length of chamber loop next to size of OMP under the same column (i.e. [Length of chamber loop/Size]) to emphasise ratios that support the direct correlation of chamber loop lengths to larger stable oligomer states.”

##This is a nice idea and we have now implemented this in full for each of the confident models in Table S3, and it (as the reviewer suggests) tallies nicely with oligomerisation state. As can be seen in Figure 6, the loops on the 8-stranded family run upwards, not outwards and so there is less correlation with oligomer breadth here, making addition of this to S4 less useful.

“Also, it may be helpful to include PISA values together with ipTM/pTM values, to indicate maximum barrel-barrel SA contact in the optimal configuration (for top few tetrameric/ pentameric/ hexameric hits).”

##Again, we thank reviewer 3 for this suggestion but note that in modelling all of the homologues, PISA values can't be used as indicators because in confident higher oligomers (e.g. hexamer), lower order oligomers (pentamer, tetramer) aren't modelled as “closed”, but as hexamers missing one/two units, hence calculation provides similar buried contacts per monomer. PISA would potentially be of use in structure prediction that uses radial closed oligomers (e.g. UniFold, SGNet) but we have stuck with the higher-performance of AlphaFold3.

“Line 370: Parr et al. propose that PopA could be transferred from predator to prey via vesicles of ~25 nm in prey periplasm (M Kaplan et al., 2023). In this manuscript, the role of PopA in the Bd predation cycle has been proposed to be involved in lipid-trapping and LPS-dispersion, likely causing prey OM-swelling, yet popA deletion in Bd has no impact on predation (line 196-197).

Is it likely that PopA has a role in the end-stage lysis of prey bacteria and predator release by contributing to membrane pliability?”

##In a prior study on cell wall modifications, we discovered that predator release was partly aided by a modified lysozyme (PMID:32968056), with the inference that there were also other factors involved. Our phenotyping of knockout strains did not reveal a progeny deficit that you would expect with compromised predator release. Of course, that is not to say it isn't a factor in environmental predation (as opposed to lab-based predation assays), but we choose not to speculate on this.

“Line 649: For EM structure deposition, PDB code provided without EMDB code.”

##Apologies for this omission, we have amended line 685 with the submission code 9GF0 for EMDB entry ID EMD-51308.

Our reply prefaced by ##### below

REVIEWERS' COMMENTS

Reviewer #1 (Remarks to the Author):

The Authors have addressed my comments - I have nothing further to add.

Many thanks for your time in reviewing this work.

Reviewer #2 (Remarks to the Author):

I appreciate the authors' hard work on this challenging protein. Although there are still many questions (to be addressed in the coming years!), I feel the added data and analysis have improved the paper a lot and am now happy to support publication.

Thanks for your time and for the appreciation of our efforts.

Reviewer #3 (Remarks to the Author):

In this revision, Parr and colleagues have improved their manuscript by supplementing new data, and have extensively addressed reviewers' comments and incorporated said recommendations. This study would be a great addition to inspire future work towards further characterisation of these novel OMP superfamilies and detailed function-mechanism of Bd0427 in predatory Bdellovibrio bacteriovorus, towards innovative biotherapeutics. Please find my unreserved support for the publication of this manuscript.

Line 185: "demonstrating high contact residency time for this ligand during simulations (Fig. 3e)" - should be Fig. 3d?

Thanks for your time and positivity during review process; thanks for spotting this error, this has now been amended.